# Numerical Composition of Differential Privacy[*]

**Sivakanth Gopi**
Microsoft Research
sigopi@microsoft.com

**Yin Tat Lee**
University of Washington
yintat@uw.edu

**Lukas Wutschitz**
Microsoft
lukas.wutschitz@microsoft.com

## Abstract

We give a fast algorithm to optimally compose privacy guarantees of differentially private (DP) algorithms to arbitrary accuracy. Our method is based on the notion of *privacy loss random variables* to quantify the privacy loss of DP algorithms. The running time and memory needed for our algorithm to approximate the privacy curve of a DP algorithm composed with itself $k$ times is $\tilde{O}(\sqrt{k})$. This improves over the best prior method by [KH21] which requires $\tilde{\Omega}(k^{1.5})$ running time. We demonstrate the utility of our algorithm by accurately computing the privacy loss of DP-SGD algorithm of Abadi et al. [ACG+16] and showing that our algorithm speeds up the privacy computations by a few orders of magnitude compared to prior work, while maintaining similar accuracy.

## 1 Introduction

Differential privacy (DP) introduced by [DMNS06] provides a provable and quantifiable guarantee of privacy when the results of an algorithm run on private data are made public. Formally, we can define an $(\varepsilon, \delta)$-differentially private algorithm as follows.

**Definition 1.1** $((\varepsilon, \delta)$-DP [DMNS06, DKM+06]). *An algorithm $\mathcal{M}$ is $(\varepsilon, \delta)$-DP if for any two neighboring databases $D, D'$ differing in exactly one user and any subset $S$ of outputs, we have $\Pr[\mathcal{M}(D) \in S] \le e^\varepsilon \Pr[\mathcal{M}(D') \in S] + \delta$.*

Intuitively, it says that looking at the outcome of $\mathcal{M}$, we cannot tell whether it was run on $D$ or $D'$. Hence an adversary cannot infer the existence of any particular user in the input database, and therefore cannot learn any personal data of any particular user.

DP algorithms have an important property called *composition*. Suppose $M_1$ and $M_2$ are DP algorithms and say $M(D) = (M_1(D), M_2(D))$, i.e., $M$ runs both the algorithms on $D$ and outputs their results. Then $M$ is also a DP algorithm.

**Proposition 1.2** (Simple composition [DKM+06, DL09]). *If $M_1$ is $(\varepsilon_1, \delta_1)$-DP and $M_2$ is $(\varepsilon_2, \delta_2)$-DP, then $M(D) = (M_1(D), M_2(D))$ is $(\varepsilon_1 + \varepsilon_2, \delta_1 + \delta_2)$-DP.*

This also holds under *adaptive composition* (denoted by $M = M_2 \circ M_1$), where $M_2$ can look at both the database and the output of $M_1$ (here $M(D) = (M_1(D), M_2(D, M_1(D)))$). It turns out that both compositions enjoy much better DP guarantees than this simple composition rule. Let $M$ be an $(\varepsilon, \delta)$-DP algorithm and let $M^{\circ k}$ denote the (adaptive) composition of $M$ with itself $k$ times. The naive composition rule shows that $M^{\circ k}$ is $(k\varepsilon, k\delta)$-DP. This was significantly improved in [DRV10].

**Proposition 1.3** (Advanced composition [DRV10, DR+14]). *If $M$ is $(\varepsilon, \delta)$-DP, then $M^{\circ k}$ is $(\varepsilon', k\delta + \delta')$-DP where*

$$\varepsilon' = \varepsilon \sqrt{2k \log\left(\frac{1}{\delta'}\right)} + k\varepsilon(e^\varepsilon - 1).$$

---

[*]Author ordering is alphabetical. Code is available at https://github.com/microsoft/prv_accountant.

Note that if $\varepsilon = O\left(\frac{1}{\sqrt{k}}\right)$ and $\delta = o\left(\frac{1}{k}\right)$, then $M^{\circ k}$ satisfies $(O_{\delta'}(1), \delta')$-DP. Using simple composition (Proposition 1.2), we can only claim that $M^{\circ k}$ is $(O(\sqrt{k}), o(1))$-DP. Thus advanced composition often results in $\sqrt{k}$-factor savings in privacy which is significant in practice. The optimal DP guarantees for $k$-fold composition of an $(\varepsilon, \delta)$-DP algorithm were finally obtained by [KOV15]. For composing different algorithms, the situation is more complicated. If $M_1, M_2, \ldots, M_k$ are DP algorithms such that $M_i$ is $(\varepsilon_i, \delta_i)$-DP, then it is shown by [MV16] that computing the *exact* DP guarantees for $M = M_1 \circ M_2 \circ \cdots \circ M_k$ is #P-complete. They also give an algorithm to approximate the DP guarantees of $M$ to desired accuracy $\eta$ which runs in

$$\tilde{O}\left(\frac{k^3 \bar{\varepsilon}(1 + \bar{\varepsilon})}{\eta}\right) \tag{1}$$

time where $\bar{\varepsilon} = (\sum_{i=1}^k \varepsilon_i)/k$.[2] If each $\varepsilon_i \approx \frac{1}{\sqrt{k}}$ (so that $M$ will satisfy reasonable privacy guarantees by advanced composition), then the running time is $\tilde{O}(k^{2.5}/\eta)$.

In most situations, DP algorithms come with a collection of $(\varepsilon, \delta)$-DP guarantees, i.e., for each value of $\varepsilon$, there exists $\delta$ such that the algorithm is $(\varepsilon, \delta)$-DP.

**Definition 1.4** (Privacy curve). *A DP algorithm $M$ is said to have privacy curve $\delta : \mathbb{R} \to [0, 1]$, if for every $\varepsilon \in \mathbb{R}$, $M$ is $(\varepsilon, \delta(\varepsilon))$-DP.*

For example the privacy curve of a Gaussian mechanism (with sensitivity 1 and noise scale $\sigma$) is given by $\delta(\varepsilon) = \Phi\left(-\varepsilon\sigma + 1/2\sigma\right) - e^\varepsilon \Phi\left(-\varepsilon\sigma - 1/2\sigma\right)$ where $\Phi(\cdot)$ is the Gaussian CDF [BW18]. Suppose we want to compose several Gaussian mechanisms, which $(\varepsilon, \delta)$-DP guarantee should we choose for each mechanism? Any choice will lead to suboptimal DP guarantees for the final composition. Instead, we need a way to compose the privacy curves directly. This was suggested through the use of privacy region in [KOV15] and explicitly studied in the $f$-DP framework of [DRS19]. $f$-DP is a dual way (and equivalent) to look at the privacy curve $\delta(\varepsilon)$.

Independently, an algorithm called *Privacy Buckets* for approximately composing privacy curves using the notion of was initiated in [MM18]. This algorithm depends on the notion of *Privacy Loss Random Variable* (PRV) [DR16], whose distribution is called *Privacy Loss Distribution* (PLD). For any DP-algorithm, one can associate a PRV and the privacy curve of that algorithm can be easily obtained from the PRV. The really useful property of PRVs is that under adaptive composition, they just add up; the PRV $Y$ of the composition $M = M_1 \circ M_2 \circ \cdots \circ M_k$ is given by $Y = \sum_{i=1}^k Y_i$ where $Y_i$ is the PRV of $M_i$.[3] Therefore, one can find the distribution of $Y$ by the convolution of the distributions of $Y_1, Y_2, \ldots, Y_k$. In an important paper, [KJH+20] proposed that one can speed up the convolutions using Fast Fourier Transform (FFT). Explicit error bounds were obtained for the approximation obtained by their algorithm in [KJH+20, KJPH21, KH21]. The running time of this algorithm was analyzed in [KH21] where it was shown that the privacy curve $\delta_M(\varepsilon)$ of $M = M_1 \circ M_2 \circ \cdots \circ M_k$ can be computed up to an additive error of $\delta_{\text{error}}$ in time

$$\tilde{O}\left(\frac{k^3 \bar{\varepsilon}}{\delta_{\text{error}}}\right), \tag{2}$$

if each algorithm $M_i$ is satisfies $(\varepsilon_i, 0)$-DP and $\bar{\varepsilon} = \frac{1}{k} \sum_{i=1}^k \varepsilon_i$. Assuming that each $\varepsilon_i \approx \frac{1}{\sqrt{k}}$, we get $\tilde{O}(k^{2.5}/\delta_{\text{error}})$ running time. Note that this is slightly worse than (1), where the denominator $\eta$ is the multiplicative error in $\delta_M$. When composing the same algorithm with itself for $k$ times, the running time can be improved to $\tilde{O}\left(\frac{k^2 \bar{\varepsilon}}{\delta_{\text{error}}}\right)$, which is $\tilde{O}\left(\frac{k^{1.5}}{\delta_{\text{error}}}\right)$ when $\bar{\varepsilon} = \frac{1}{\sqrt{k}}$.

**Moments Accountant and Renyi DP**   In an influential paper where they introduce Differentially Private Deep Learning, [ACG+16] proposed a method called the Moments Accountant (MA) for giving an upper bound the privacy curve of a composition of DP algorithms. They applied their method to bound the privacy loss of differentially private Stochastic Gradient Descent (DP-SGD) algorithm which they introduced. Analyzing the privacy loss of DP-SGD involves composing the

---

[2]$\varepsilon$ has an additive error of $\eta$ and $\delta$ has a multiplicative error of $\eta$.

[3][KJH+20] only state this for non-adaptive composition. In this paper we show how to extend this to adaptive composition as well.

privacy curve of each iteration of training with itself $k$ times, where $k$ is the total of number of training iterations. Typical values of $k$ range from 1000 to 300000 (such as when training large models like GPT3). The Moments Accountant was subsumed into the framework of Renyi Differential Privacy (RDP) introduced by [Mir17]. The running time of these accountants are independent of $k$, but they only give an upper bound and cannot approximate the privacy curve to arbitrary accuracy.

DP-SGD is one of the most important DP algorithms in practice, because one can use it to train neural networks to achieve good privacy-vs-utility tradeoffs. Therefore obtaining accurate and tight privacy guarantees for DP-SGD is important. For example reducing $\varepsilon$ from 2 to 1, can mean that one can train the network for 4 times more epochs while staying within the same privacy budget. Therefore DP-SGD is one of the main motivations for this work.

There are also situations when the PRVs do not have bounded moments and so Moments Accountant or Renyi DP cannot be applied for analyzing privacy. An example of such an algorithm is the DP-SGD-JL algorithm of [BGK$^+$21] which uses numerical composition of PRVs to analyze privacy.

**GDP Accountant** [DRS19, BDLS19] introduced the notion of Gaussian Differential Privacy (GDP) and used it to develop an accountant for DP-SGD. The accountant is based on central limit theorem and only gives an approximation to the true privacy curve, where the approximation gets better with $k$. But as we show in Figure 1, GDP accountant can significantly underreport the true epsilon value.

Several different notions of privacy were introduced for obtaining good upper bounds on the privacy curve of composition of DP algorithms such as Concentrated DP (CDP) [DR16, BS16], Truncated CDP [BDRS18] etc. None of these methods can approximate the privacy curve of compositions to arbitrary accuracy. The notion of $f$-DP introduced by [DRS19], allows for a lossless composition theorem, but computing the privacy curve of composition seems computationally hard and they do not give any algorithms for doing it.

## 1.1 Our Contributions

The main contribution of this work is a new algorithm with an improved analysis for computing the privacy curve of the composition of a large number of DP algorithms.

**Theorem 1.5** (Informal version of Theorem 5.5)**.** *Suppose $M_1, M_2, \ldots, M_k$ are DP algorithms. Then the privacy curve $\delta_M(\varepsilon)$ of adaptive composition $M = M_1 \circ M_2 \circ \cdots \circ M_k$ can be approximated in time*

$$O\left( \frac{\varepsilon_{\mathrm{upper}}\, k^{1.5} \log k \sqrt{\log \frac{1}{\delta_{\mathrm{error}}}}}{\varepsilon_{\mathrm{error}}} \right), \tag{3}$$

*where $\varepsilon_{\mathrm{error}}$ is the additive error in $\varepsilon$, $\delta_{\mathrm{error}}$ is the additive error in $\delta$ and $\varepsilon_{\mathrm{upper}}$ is an upper bound on $\max\left\{ \varepsilon_M(\delta_{\mathrm{error}}), \max_i \varepsilon_{M_i}\left(\frac{\delta_{\mathrm{error}}}{k}\right) \right\}$.*[4]

If each $M_i$ satisfies $\left( \frac{1}{\sqrt{k}}, \frac{o(1)}{k} \right)$-DP, then by advanced composition (Proposition 1.3), we can set $\varepsilon_{\mathrm{upper}} = O(1)$. Therefore the running time of our algorithm in this case is $\tilde{O}\left( \frac{k^{1.5} \sqrt{\log \frac{1}{\delta_{\mathrm{error}}}}}{\varepsilon_{\mathrm{error}}} \right)$. We can save a factor of $k$, when we compose the same algorithm with itself $k$ times.

**Theorem 1.6.** *Suppose $M$ is a DP algorithm. Then the privacy curve $\delta_{M^{\circ k}}(\varepsilon)$ of $M$ (adaptively) composed with itself $k$ times can be approximated in time*

$$O\left( \frac{\varepsilon_{\mathrm{upper}}\, k^{\frac{1}{2}} \log k \sqrt{\log \frac{1}{\delta_{\mathrm{error}}}}}{\varepsilon_{\mathrm{error}}} \right), \tag{4}$$

*where $\varepsilon_{\mathrm{error}}$ is the additive error in $\varepsilon$, $\delta_{\mathrm{error}}$ is the additive error in $\delta$ and $\varepsilon_{\mathrm{upper}}$ is an upper bound on $\max\left\{ \varepsilon_{M^{\circ k}}(\delta_{\mathrm{error}}), \varepsilon_M\left(\frac{\delta_{\mathrm{error}}}{k}\right) \right\}$.*

Thus we improve the state-of-the-art by at least a factor of $k$ in running time. We also note that our algorithm improves the memory required by a factor of $k$. See Figure 1 for a comparison of

---

[4] $\varepsilon_M(\delta)$ is the inverse of $\delta_M(\varepsilon)$.

our algorithm with that of [KJPH21]. Also note that RDP Accountant (equivalent to the Moments Accountant) significantly overestimates the true $\varepsilon$, while the GDP Accountant significantly underestimates the true $\varepsilon$. In contrast, the upper and lower bounds provided by our algorithm lie very close to each other.

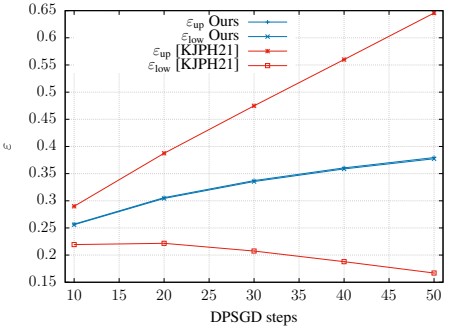

(a) Our algorithm gives much closer upper and lower bounds on the true privacy curve compared to [KJPH21], under the same mesh size of $4 \times 10^{-5}$. Our upper and lower bounds are nearly coinciding.

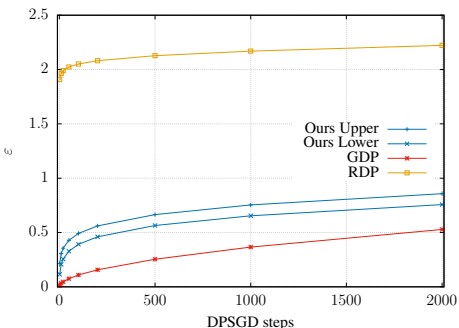

(b) Our algorithm can improve significantly over the RDP Accountant. We also see that GDP Accountant can significantly underreport the true $\varepsilon$. We have set $\varepsilon_{\text{error}} = 0.1$, $\delta_{\text{error}} = \delta/1000$ here.

Figure 1: Case study on DP-SGD. Sampling probability $p = 10^{-3}$, noise scale $\sigma = 0.8$, $\delta = 10^{-7}$.

**Our Techniques** Our algorithm (also the prior work of [KJH+20]) proceeds by approximating the privacy loss random variables (PRVs) by truncating and discretizing them. We then use Fast Fourier Transform (FFT) to convolve the distributions efficiently. The main difference is in the approximation procedure and the error analysis. In the approximation procedure, we correct the approximation so that the expected value of the discretization matches with the expected value of the PRV.

To analyze the approximation error, we introduce the concept of *coupling approximation* (Definition 5.1), which is a variant of Wasserstein (optimal transport) distance specifically tailored to this application. We first show that the approximation output by our algorithm to each privacy random variable is a good coupling approximation. We then show that when independent coupling approximations are added, cancellation happens between the errors due to Hoeffding bound, producing a much better coupling approximation than one naively expects from the triangle inequality. This allows us to choose the mesh size in our discretization to be $\approx \frac{1}{\sqrt{k}}$, whereas [KH21] choose a mesh size of $\approx \frac{1}{k}$. The other improvement is the truncation procedure. We give a tight tail bound of the PRVs (Lemma 5.4). This allows us to choose the domain size for in truncation to be $\approx \tilde{O}(1)$, whereas [KH21] choose $\approx \tilde{O}(\sqrt{k})$. Both ideas together saves a factor of $k$ in the run time and memory.

For the analysis, the previous paper analyzes the discretization error by studying the stability of convolution. This leads to complicated calculations with the runtime linear in $1/\delta_{\text{error}}$ (see (2)). Since $\delta_{\text{error}} \ll \delta \ll 1/N$ is required to give meaningful privacy guarantee ($N$ is the number of users), this term $1/\delta_{\text{error}}$ is huge. In this paper, we show various facts about how coupling approximation accumulates and use them to give a runtime depending only on $\sqrt{\log(1/\delta_{\text{error}})}$.

## 2 DP Preliminaries

Given a DP algorithm $\mathcal{M}$, for each value of $\varepsilon \geq 0$, there exists some $\delta \in [0, 1]$ such that $\mathcal{M}$ is $(\varepsilon, \delta)$-DP. We can represent all these privacy guarantees by a function $\delta_{\mathcal{M}}(\varepsilon) : \mathbb{R}^{\geq 0} \rightarrow [0, 1]$ and say that $\delta_{\mathcal{M}}(\cdot)$ is the *privacy curve* of $\mathcal{M}$. This inspires the following definition of a privacy curve between two random variables.

**Definition 2.1** (Privacy curve). *Given two random variables $X, Y$ supported on some set $\Omega$, define $\delta(X||Y) : \mathbb{R} \rightarrow [0, 1]$ as:*

$$\delta(X||Y)(\varepsilon) = \sup_{S \subset \Omega} \Pr[Y \in S] - e^{\varepsilon} \Pr[X \in S].$$

Therefore an algorithm $\mathcal{M}$ is $(\varepsilon, \delta)$-DP iff $\delta\left(\mathcal{M}(D)||\mathcal{M}(D')\right)(\varepsilon) \leq \delta$ for all neighboring databases $D, D'$.

**Remark 2.2.** *Note that not all functions $\delta : \mathbb{R} \to [0, 1]$ are privacy curves. A characterization of privacy curves can be obtained using the $f$-DP framework of [DRS19]. The notion of privacy curve $\delta(X||Y)$ and tradeoff function $T(X||Y)$ are dual to each other via convex duality [DRS19]. This implies a characterization of privacy curves as shown in [ZDW21].*

**Definition 2.3** (Composition of privacy curves [DRS19]). *Let $\delta_1 \equiv \delta(X_1||Y_1)$ and $\delta_2 \equiv \delta(X_2||Y_2)$ be any two privacy curves. The composition of the privacy curves, denoted by $\delta_1 \otimes \delta_2$, is defined as*

$$\delta_1 \otimes \delta_2 \equiv \delta\left((X_1, X_2)||(Y_1, Y_2)\right)$$

*where $X_1, X_2$ are independently sampled and $Y_1, Y_2$ are independently sampled.*

Note that there can be many pairs of random variables which have the same privacy curve, but the above operation is well-defined. If $\delta(X_1||Y_1) \equiv \delta(X_1'||Y_1')$ and $\delta(X_2||Y_2) \equiv \delta(X_2'||Y_2')$, then it was shown by [DRS19] that

$$\delta\left((X_1, X_2)||(Y_1, Y_2)\right) = \delta\left((X_1', X_2')||(Y_1', Y_2')\right).$$

[DRS19] also show that $\otimes$ is a commutative and associative operation.

Given two DP algorithms $M_1$ and $M_2$, the adaptive composition $(M_2 \circ M_1)(D)$ is an algorithm which outputs $(M_1(D), M_2(D, M_1(D)))$, i.e., $M_2$ can look at the database $D$ and also the output of the previous algorithm $M_1(D)$. Adaptive composition of more than two algorithms is similarly defined. Suppose $M_1$ has privacy curve $\delta_1$ and $M_2$ has privacy curve $\delta_2$ (i.e., $M_2(\cdot, y)$ is a DP algorithm with privacy curve $\delta_2$ for any fixed $y$.). The following composition theorem shows how to get the privacy curve of $M_2 \circ M_1$.

**Theorem 2.4** (Composition theorem [DRS19]). *Let $M_1, M_2, \ldots, M_k$ be DP algorithms with privacy curves given by $\delta_1, \delta_2, \ldots, \delta_k$ respectively. The privacy curve of the adaptive composition $M_k \circ M_{k-1} \circ \cdots \circ M_1$ is given by $\delta_1 \otimes \delta_2 \otimes \cdots \otimes \delta_k$.*

# 3 Privacy loss random variables

The notion of *privacy loss random variables* (PRVs) is a unique way to assign a pair $(X, Y)$ for any privacy curve $\delta$ such that $\delta \equiv \delta(X||Y)$. PRVs allow us to compute composition of two algorithms via summing random variables (Theorem 3.5) (equivalently, convolving their distributions). Thus PRVs can be thought of as a *reparametrization of privacy curves* where composition becomes convolution. In this paper, we differ from the usual definition of PRVs given in [DR16, KJH+20], which are tied to a specific algorithm. Instead we think of them as a reparametrization of privacy curves and study them directly. This allows us to succinctly prove many useful properties of PRVs.

Let $\overline{\mathbb{R}} = \mathbb{R} \cup \{-\infty, \infty\}$ be the extended real line where we define $\infty + x = \infty$ and $-\infty + x = -\infty$ for $x \in \mathbb{R}$.

**Definition 3.1** (Privacy loss random variables (PRVs)). *Given a privacy curve $\delta : \mathbb{R} \to [0, 1]$, we say that $(X, Y)$ are privacy loss random variables for $\delta$, if they satisfy the following conditions:*

- *$X, Y$ are supported on $\overline{\mathbb{R}}$,*

- *$\delta(X||Y) \equiv \delta$,*

- *$Y(t) = e^t X(t)$ for every $t \in \mathbb{R}$ and*

- *$Y(-\infty) = 0$ and $X(\infty) = 0$*

*where $X(t), Y(t)$ are probability density functions of $X, Y$ respectively.*

Mathematically, the correct way to write the condition $Y(t) = e^t X(t)$ is to say that $\mathbb{E}_Y[\phi(Y)] = \mathbb{E}_X[\phi(X)e^X]$ for all test functions $\phi : \overline{\mathbb{R}} \to [0, 1]$ with $\phi(\infty) = \phi(-\infty) = 0$. This will generalize to all situations where $X, Y$ are continuous or discrete or both. For ease of exposition, we ignore this complication and assume that $X(t), Y(t)$ represent the PDFs if $X, Y$ are continuous at $t$, or the probability masses if they have point masses at $t$.

The following theorem shows that the PRVs for a privacy curve $\delta = \delta(P||Q)$ are given by the log-likelihood random variables of $P, Q$.

**Theorem 3.2.** *Let $\delta : \mathbb{R} \to [0,1]$ be a privacy curve given by $\delta \equiv \delta(P\|Q)$ where $P, Q$ are two random variables supported on $\Omega$. The PRVs $(X, Y)$ for the privacy curve $\delta$ are given by[5]:*

$$X = \log\left(\frac{Q(\omega)}{P(\omega)}\right) \text{ where } \omega \sim P,$$

$$Y = \log\left(\frac{Q(\omega)}{P(\omega)}\right) \text{ where } \omega \sim Q.$$

The following theorem provides a formula for computing the privacy curve $\delta$ in terms of the PRVs and conversely a formula for PRVs in terms of the privacy curve. A similar statement appears in [SMM19, KJH+20].

**Theorem 3.3.** *The privacy curve $\delta$ can be expressed in terms of PRVs $(X, Y)$ as:*

$$\delta(\varepsilon) = \Pr[Y > \varepsilon] - e^\varepsilon \Pr[X > \varepsilon] = \mathbb{E}_Y[(1 - e^{\varepsilon-Y})_+] = \Pr[Y \geq \varepsilon + Z]. \tag{5}$$

*where $Z$ is an exponential random variable.[6] Conversely, given a privacy curve $\delta : \mathbb{R} \to [0,1]$, we can compute the PDFs of its PRVs $(X, Y)$ as:*

$$Y(t) = \delta''(t) - \delta'(t) \text{ and } X(t) = e^t(\delta''(t) - \delta'(t)). \tag{6}$$

**Remark 3.4.** *Theorem 3.3 shows that the PRVs $X, Y$ do not depend on the particular $P, Q$ used to represent the privacy curve $\delta$ in Theorem 3.2. So we should think of the PDF of of the PRV $Y$ (or $X$) as an equivalent reparametrization of the privacy curve $\delta : \mathbb{R} \to [0,1]$, just as the notion of $f$-DP [DRS19] is a reparametrization of the privacy curve $\delta$.*

PRVs are useful in computing privacy curves because the composition of two privacy curves can be computed by adding the corresponding pairs of PRVs. A similar statement appears in [DR16].

**Theorem 3.5.** *Let $\delta_1, \delta_2$ be two privacy curves with PRVs $(X_1, Y_1)$ and $(X_2, Y_2)$ respectively. Then the PRVs for $\delta_1 \otimes \delta_2 = \delta(X_1, X_2\|Y_1, Y_2)$ are given by $(X_1 + X_2, Y_1 + Y_2)$. In particular,*

$$\delta_1 \otimes \delta_2 = \delta(X_1 + X_2\|Y_1 + Y_2).$$

*Proof.* Let $(X, Y)$ be the privacy random variables for $\delta(X_1, X_2\|Y_1, Y_2)$. By Theorem 3.2,

$$
\begin{aligned}
X &= \log\left(\frac{(Y_1, Y_2)(t_1, t_2)}{(X_1, X_2)(t_1, t_2)}\right) \text{ where } (t_1, t_2) \sim (X_1, X_2) \\
&= \log\left(\frac{Y_1(t_1)Y_2(t_2)}{X_1(t_1)X_2(t_2)}\right) \text{ where } t_1 \sim X_1, t_2 \sim X_2 \\
&\qquad\qquad \text{(By independence of } X_1, X_2 \text{ and indpendence of } Y_1, Y_2) \\
&= \log\left(e^{t_1} \cdot e^{t_2}\right) \text{ where } t_1 \sim X_1, t_2 \sim X_2 \\
&= t_1 + t_2 \text{ where } t_1 \sim X_1, t_2 \sim X_2 \\
&= X_1 + X_2.
\end{aligned}
$$

Similarly,

$$
\begin{aligned}
Y &= \log\left(\frac{(Y_1, Y_2)(t_1, t_2)}{(X_1, X_2)(t_1, t_2)}\right) \text{ where } (t_1, t_2) \sim (Y_1, Y_2) \\
&= t_1 + t_2 \text{ where } t_1 \sim Y_1, t_2 \sim Y_2 \\
&= Y_1 + Y_2.
\end{aligned}
$$

$\square$

In the supplementary material, we provide a proof of Theorems 3.2 and 3.3. We also discuss how to compute the PRVs for a subsampled mechanism given the PRVs for the original mechanism and give examples of PRVs for few standard mechanisms. These are used in our experiments to calculate the PRVs for DP-SGD.

---

[5]Here $Q(\omega)$ and $P(\omega)$ are the probability density functions of $Q, P$ respectively. Note that the mathematically precise way is to replace the ratio $\frac{Q(\omega)}{P(\omega)}$ by the Radon-Nikodym derivative $\frac{dQ}{dP}(\omega)$.

[6]For $x \in \mathbb{R}$, $x_+ = \max\{x, 0\}$.

## 4 Numerical composition of privacy curves

In this section, we present an efficient and numerically accurate method, ComposePRV (Algorithm 1), for composing privacy guarantees by utilizing the notion of PRVs.

---

**Algorithm 1:** ComposePRV: Composing privacy curves using PRVs

---

**Input:** CDFs of PRVs $Y_1, Y_2, \ldots, Y_k$, mesh size $h$, Truncation parameter $L \in \frac{h}{2} + h\mathbb{Z}^{>0}$

**Output:** PDF of an approximation $\widetilde{Y}$ for $Y = \sum_{i=1}^{k} Y_i$. $\widetilde{Y}$ will be supported on
$\quad \mu + (h\mathbb{Z} \cap [-L, L])$ for some $\mu \in [0, \frac{h}{2}]$.

**for** $\ell = 1$ *to* $k$ **do**
$\quad \mid \quad \widetilde{Y}_i \leftarrow \mathsf{DiscretizePRV}(Y_i, L, h)$;
**end**

Compute PDF Of $\widetilde{Y} = \widetilde{Y}_1 \oplus_L \widetilde{Y}_2 \oplus_L \cdots \oplus_L \widetilde{Y}_k$ by convolving PDFs of $\widetilde{Y}_1, \widetilde{Y}_2, \ldots, \widetilde{Y}_k$ using FFT;

Compute $\delta_{\widetilde{Y}}(\varepsilon) = \mathbb{E}_{\widetilde{Y}} \left[ \left(1 - e^{\varepsilon - \widetilde{Y}}\right)_+ \right]$ for all $\varepsilon \in [0, L]$;

Return $\widetilde{Y}, \delta_{\widetilde{Y}}(\cdot)$

---

In the algorithm ComposePRV, we compute the circular convolution $\oplus_L$ using Fast Fourier Transform (FFT). Fix some $L > 0$. For $x \in \mathbb{R}$, we define $x \pmod{2L} = x - 2Ln$ where $n \in \mathbb{Z}$ is chosen such that $x - 2Ln \in (-L, L]$. Given $x, y$, we define the circular addition $x \oplus_L y = x + y \pmod{2L}$. When we use FFT to compute the convolution of two discrete distributions $Y_1, Y_2$ supported on $h\mathbb{Z} \cap [-L, L]$, we are implicitly calculating the the distribution of $Y_1 \oplus_L Y_2$. We will later show that $\widetilde{Y}_1 \oplus_L \widetilde{Y}_2 \oplus_L \cdots \oplus_L \widetilde{Y}_k$ is a good approximation of $Y_1 + Y_2 + \cdots + Y_k$.

The subroutine DiscretizePRV (Algorithm 2) is used to truncate and discretize PRVs. In this subroutine, we shift the discretized random variables such that it has the same mean as the original variables. This is one of main differences between our algorithm and the algorithm in [KJPH21, KH21]. We show that this significantly decreases the discretization error and allow us to use much coarser mesh $h \approx 1/\sqrt{k}$ instead of $h \approx 1/k$.

---

**Algorithm 2:** DiscretizePRV: Discretize and truncate a PRV

---

**Input:** $\mathrm{CDF}_Y(\cdot)$ of a PRV $Y$, mesh size $h$, Truncation parameter $L \in \frac{h}{2} + h\mathbb{Z}^{>0}$

**Output:** PDF of an approximation $\widetilde{Y}$ supported on $\mu + (h\mathbb{Z} \cap [-L, L])$ for some $\mu \in [0, \frac{h}{2}]$.

$n \leftarrow \frac{L - \frac{h}{2}}{h}$;
**for** $i = -n$ *to* $n$ **do**
$\quad \mid \quad q_i \leftarrow \mathrm{CDF}_Y(ih + h/2) - \mathrm{CDF}_Y(ih - h/2)$;
**end**
$q \leftarrow q / \left(\sum_{i=-n}^{n} q_i\right)$;      // Normalize $q$ to make it a probability distribution
$Y^L \leftarrow Y\big|_{|Y| \leq L}$ (i.e., $Y$ conditioned on $|Y| \leq L$);
$\mu \leftarrow \mathbb{E}[Y^L] - \sum_{i=-n}^{n} ih \cdot q_i$;
$\widetilde{Y} \leftarrow \big\{ ih + \mu \quad \text{w.p. } q_i \text{ for } -n \leq i \leq n$;
Return $\widetilde{Y}$;

---

For simplicity, throughout this paper, we will assume that the PRVs $Y_1, Y_2, \ldots, Y_k$ do not have any mass at $\infty$. This is with out loss of generality. Suppose $\Pr[Y_i = \infty] = \delta_i$ for each $i$. Let $Y_i' = Y_i|_{Y_i \neq \infty}$. Then

$$Y_1 + Y_2 + \cdots + Y_k = \begin{cases} Y_1' + Y_2' + \cdots + Y_k' & \text{w.p. } (1 - \delta_1)(1 - \delta_2) \cdots (1 - \delta_k) \\ \infty & \text{w.p. } 1 - (1 - \delta_1)(1 - \delta_2) \cdots (1 - \delta_k). \end{cases}$$

Therefore we can use Algorithm 1 to approximate the distribution of $Y_1' + Y_2' + \cdots + Y_k'$, and use it to approximate the distribution of $Y_1 + Y_2 + \cdots + Y_k$.

# 5 Error analysis

To analyze the discretization error, we introduce the notion of *coupling approximation*, a variant of Wasserstein distance. Intuitively, a good coupling approximation is a coupling where the two random variables are close to each other with high probability.

**Definition 5.1** (coupling approximation). *Given two random variables $Y_1, Y_2$, we write $|Y_1 - Y_2| \leq_\eta h$ if there exists a coupling between $Y_1, Y_2$ such that $\Pr[|Y_1 - Y_2| > h] \leq \eta$.*

The following lemma shows that if we have a good coupling approximation $\widetilde{Y}$ to a PRV $Y$, then the privacy curves $\delta_Y(\varepsilon)$ and $\delta_{\widetilde{Y}}(\varepsilon)$ should be close.

**Lemma 5.2.** *If $Y$ and $\widetilde{Y}$ are two random variables such that $|Y - \widetilde{Y}| \leq_\eta h$, then for every $\varepsilon \in \mathbb{R}$,*

$$\delta_{\widetilde{Y}}(\varepsilon + h) - \eta \leq \delta_Y(\varepsilon) \leq \delta_{\widetilde{Y}}(\varepsilon - h) + \eta.$$

*Proof.* By Theorem 3.2, $\delta_Y(\varepsilon) = \Pr[Y \geq \varepsilon + Z]$ and hence

$$
\begin{aligned}
\delta_Y(\varepsilon) &= \Pr[Y - \widetilde{Y} + \widetilde{Y} \geq \varepsilon + Z] \\
&\leq \Pr[Y - \widetilde{Y} \geq h] + \Pr[\widetilde{Y} \geq \varepsilon - h + Z] \\
&\leq \eta + \delta_{\widetilde{Y}}(\varepsilon - h).
\end{aligned}
$$

Similarly, we have $\delta_{\widetilde{Y}}(\varepsilon) \leq \eta + \delta_Y(\varepsilon - h)$ for all $\varepsilon \in \mathbb{R}$. $\qquad\square$

Therefore the goal of our analysis is to show that the ComposePRV algorithm finds a good coupling approximation $\widetilde{Y}$ to $Y = \sum_{i=1}^k Y_i$. We first show that the DiscretizePRV algorithm computes a good coupling approximation to the PRVs and crucially, it preserves the expected value after truncation. Lemma D.5 shows that $|\widetilde{Y} - Y^L| \leq_0 h$ where $\widetilde{Y}$ is the approximation of a PRV $Y$ output by Algorithm 2 and $Y^L$ is the truncation of $Y$ to $[-L, L]$.

We then use the following key lemma which shows that when we add independent coupling approximations (where expected values match), we get a much better coupling approximation than what the triangle inequality predicts.

**Lemma 5.3.** *Suppose $Y_1, Y_2, \ldots, Y_k$ and $\widetilde{Y}_1, \widetilde{Y}_2, \ldots, \widetilde{Y}_k$ are two collections of independent random variables such that $|Y_i - \widetilde{Y}_i| \leq_0 h$ and $\mathbb{E}[Y_i] = \mathbb{E}[\widetilde{Y}_i]$ for all $i$, then*

$$\left| \sum_{i=1}^k Y_i - \sum_{i=1}^k \widetilde{Y}_i \right| \leq_\eta h \sqrt{2k \log \frac{2}{\eta}}.$$

*Proof.* Let $X_i = Y_i - \widetilde{Y}_i$ where $(Y_i, \widetilde{Y}_i)$ are coupled such that $|Y_i - \widetilde{Y}_i| \leq h$ w.p. 1. Then $X_i \in [-h, h]$ w.p. 1. Note that $X_1, X_2, \ldots, X_k$ are independent of each other. By Hoeffding's inequality,

$$\Pr\left[ \left| \sum_i X_i \right| \geq t \right] \leq 2 \exp\left( -\frac{2t^2}{k(2h)^2} \right) = \eta$$

if we set $t = h \sqrt{2k \log \frac{2}{\eta}}$. $\qquad\square$

This lemma shows that the error of $k$ times composition is around $\sqrt{k} \cdot h$ and hence setting $h \approx 1/\sqrt{k}$ gives small enough error. Next, we bound the domain size $L$. Naively, the domain size $L$ should be of the order of $\sqrt{k}$ because $Y$ is the sum of $k$ independent random variables with each bounded by a constant. In the supplementary material, we prove a tighter tail bound of $Y$.

**Lemma 5.4.** *Let $(X, Y)$ be the privacy random variables for a $(\varepsilon, \delta)$-DP algorithm, then for any $t \geq 0$, we have*

$$\Pr[|Y| \geq \varepsilon + t] \leq \frac{\delta(1 + e^{-\varepsilon - t})}{1 - e^{-t}}.$$

This shows that $\Pr[|Y| \geq \varepsilon + 2] \leq \frac{4}{3}\delta$ and hence truncating the domain with $L = 2 + \varepsilon$ only introduces an additive $\delta$ error in the privacy curve. Therefore, if the composition satisfies a good privacy guarantee (namely $\varepsilon = O(1)$ for small enough $\delta$), we can truncate the domain at $L = \Theta(1)$. Together with the fact that mesh size is $1/\sqrt{k}$, this gives a $O(\sqrt{k})$-time algorithm for computing the privacy curve when we compose the same mechanism with itself $k$ times. The following theorem gives a formal statement of the error bounds of our algorithm, it is proved in the supplementary material.

**Theorem 5.5.** *Let $\varepsilon_{\text{error}}, \delta_{\text{error}} > 0$ be some fixed error terms. Let $\mathcal{M}_1, \mathcal{M}_2, \ldots, \mathcal{M}_k$ be DP algorithms with privacy curves $\delta_{\mathcal{M}_i}(\varepsilon)$. Let $Y_i$ be the PRV corresponding to $\mathcal{M}_i$ such that $\delta_{\mathcal{M}_i}(\varepsilon) = \delta_{Y_i}(\varepsilon)$ for $\varepsilon \geq 0$. Let $\mathcal{M}$ be the (adaptive) composition of $\mathcal{M}_1, \mathcal{M}_2, \ldots, \mathcal{M}_k$ and let $\delta_{\mathcal{M}}(\varepsilon)$ be its privacy curve. Set $L \geq 2 + \varepsilon_{\text{error}}$ sufficiently large such that*

$$\sum_{i=1}^{k} \delta_{\mathcal{M}_i}(L-2) \leq \frac{\delta_{\text{error}}}{8} \text{ and } \delta_{\mathcal{M}}(L - 2 - \varepsilon_{\text{error}}) \leq \frac{\delta_{\text{error}}}{4}. \tag{7}$$

*Let $\widetilde{Y}$ be the approximation of $Y = \sum_{i=1}^{k} Y_i$ produced by* **ComposePRV** *algorithm with mesh size*

$$h = \frac{\varepsilon_{\text{error}}}{\sqrt{\frac{k}{2} \log \frac{12}{\delta_{\text{error}}}}}.$$

*Then*

$$\delta_{\widetilde{Y}}(\varepsilon + \varepsilon_{\text{error}}) - \delta_{\text{error}} \leq \delta_Y(\varepsilon) = \delta_{\mathcal{M}}(\varepsilon) \leq \delta_{\widetilde{Y}}(\varepsilon - \varepsilon_{\text{error}}) + \delta_{\text{error}}. \tag{8}$$

*Furthermore, our algorithm takes $O\left(b\frac{L}{h} \log\left(\frac{L}{h}\right)\right)$ time where $b$ is the number of distinct algorithms among $\mathcal{M}_1, \mathcal{M}_2, \ldots, \mathcal{M}_k$.*

**Remark 5.6.** *A simple way to set $L$ such that the condition (7) holds is by choosing an $L$ such that:*

$$L \geq 2 + \max\left\{\varepsilon_{\text{error}} + \varepsilon_{\mathcal{M}}\left(\frac{\delta_{\text{error}}}{4}\right), \max_{i \in [k]} \varepsilon_{\mathcal{M}_i}\left(\frac{\delta_{\text{error}}}{8k}\right)\right\} \tag{9}$$

*where $\varepsilon_{\mathcal{A}}(\delta)$ is the inverse of $\delta_{\mathcal{A}}(\varepsilon)$. To set the value of $L$, we do not need the exact value of $\varepsilon_{\mathcal{M}}$ (or $\varepsilon_{\mathcal{M}_i}$). We only need an upper bound on $\varepsilon_{\mathcal{M}}$, which can often by obtained by using the RDP Accountant or any other method to derive upper bounds on privacy.*

## 6 Experiments

We demonstrate the utility of our composition method by computing the privacy curves for the DP-SGD algorithm, which is one of the most important algorithms in differential privacy.

The DP-SGD algorithm [ACG+16] is a variant of stochastic gradient descent with $k$ steps. In each step, the algorithm selects a $p$ fraction of training examples uniformly at random. The algorithm adds a Gaussian vector with variance $\propto \sigma^2$ to the clipped gradient of the selected batch. Then it performs a gradient step (or any other iterative methods) using the noisy gradient computed. The privacy loss of DP-SGD involves composing the privacy curve of each iteration with itself $k$ times. The PRVs for each iteration have a closed form and depend only $p, \sigma$ (see supplementary material). Our algorithms use this closed form of PRVs.

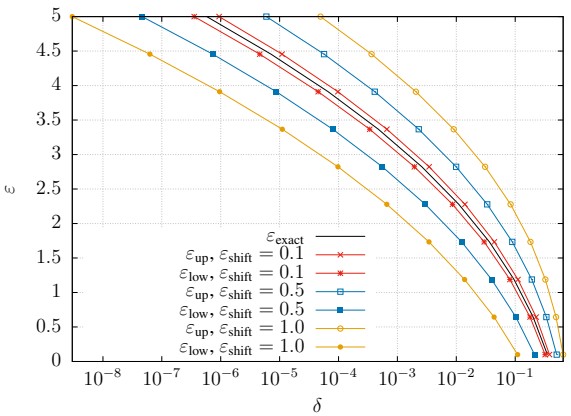

Figure 2: Setting $p = 1$ and comparing to the analytical solution (10).

See Figure 1(b) for the comparison between our algorithm and the GDP and RDP Accountant. Our method provides a lower and upper bound of the privacy curve according to (8). In Figure 1(a), we compare our algorithm with [KJPH21]

(implemented in [KP21]). Under the same mesh size, our algorithm computes a much closer upper and lower bound.

We validate our program for the case $p = 1$. When $p = 1$, we have an exact formula for

$$\delta(\varepsilon) = \Phi\left(-\frac{\varepsilon}{\mu} + \frac{\mu}{2}\right) - e^{\varepsilon}\Phi\left(-\frac{\varepsilon}{\mu} - \frac{\mu}{2}\right) \tag{10}$$

where $\mu = \frac{\sqrt{k}}{\sigma}$. In Figure 2, we show that the true privacy curve is indeed sandwiched between the bounds we compute and that the vertical distance between our bounds is indeed $2\varepsilon_{\text{error}}$ with a neglible $\delta_{\text{error}}$ of $10^{-10}$.

**Floating point errors**   Note that our error analysis in Section 5 ignores floating point errors. This is because they are negligible compared to the discretization and truncation errors we analyzed in Section 5 for the range of $\delta$ we are interested in. See supplementary material for more details.

### 6.1 Comparison with [KJPH21]

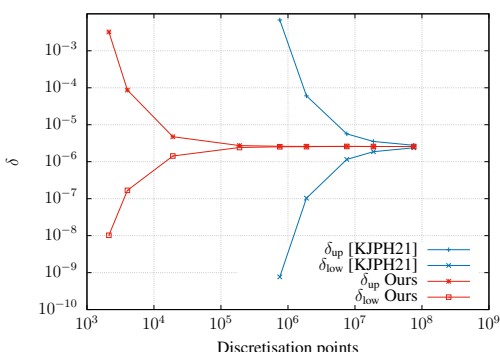 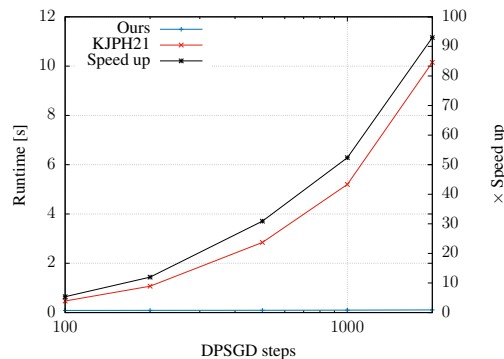

Figure 3: (a) Comparison of error bounds of $\delta$ with varying number of discretisation points for $p = 4 \times 10^{-3}, \sigma = 0.8, \varepsilon = 1.5, k = 1000$. (b) Comparing runtimes for our algorithm with that of [KJPH21] when aligned on accuracy for $\sigma = 0.8, p = 4 \times 10^{-3}$. We can see a significant reduction in runtime in particular for large number of DPSGD steps. We were not able to run the algorithm of [KJPH21] beyond 2,000 steps, since it becomes unstable beyond that point.[7]. We also plot the speed up directly on the secondary $y$-axis.

In this section, we provide more results demonstrating the practical use of our algorithm. We compare runtimes of our algorithm with [KJPH21], which is the state-of-the-art, for typical values of privacy parameters ($\sigma = 0.8$, $p = 4 \times 10^{-3}$, $\varepsilon = 1.5$). See Figure 3(a) for the effect of the number of discretisation points $n$ on the accuracy of $\delta$. Our algorithm requires about a few orders of magnitude smaller number of discretization points to converge compared to the algorithm of [KJPH21].

In order to compare runtimes, we align the accuracy of both FFT algorithms. We set numerical parameters (number of discretization bins and domain length) such that both algorithms give similarly accurate bounds and verify it visually. Figure 3(b) shows a significant speedup (100x) using our algorithms. We note that runtimes are directly proportional to the memory required by the algorithms and so a separate memory analysis is not required; the runtime and memory are dominated by the number of points in the discretization of PRV. All experiments are performed on a Intel Xeon W-2155 CPU with 3.30GHz with 128GB of memory.

## Acknowledgments and Disclosure of Funding

We would like to thank Janardhan Kulkarni and Sergey Yekhanin for several useful discussions and encouraging us to work on this problem. L.W. would like to thank Daniel Jones and Victor Rühle for fruitful discussions and helpful guidance.

---

[7]We are using the implementation of [KJPH21] from [KP21].

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
