(see Figure 4(b)). Figure 4(a) illustrates the runtimes for varying numbers of DPSGD steps. We observe a significant reduction in the runtime using our algorithms.

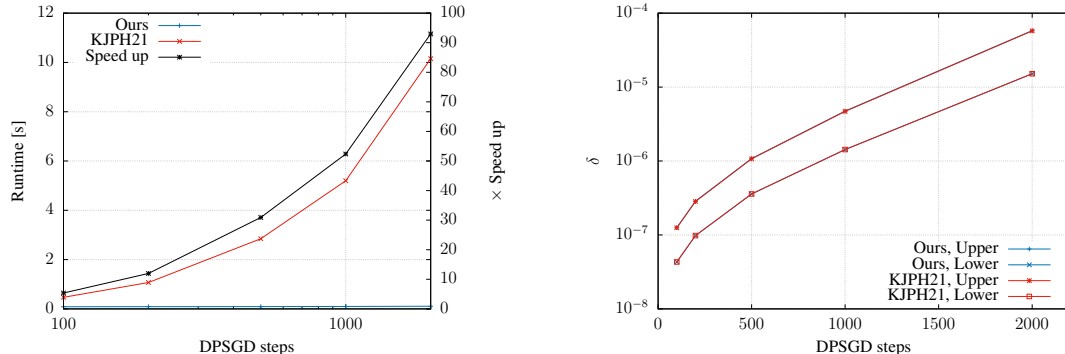

Figure 4: (a) Comparing runtimes for our algorithm with that of [KJPH21] when aligned on accuracy for $\sigma = 0.8$, $p = 4 \times 10^{-3}$. We can see a significant reduction in runtime in particular for large number of DPSGD steps. We were not able to run the algorithm of [KJPH21] beyond 2,000 steps, since it becomes unstable beyond that point.[8]. We also plot the speed up directly on the secondary $y$-axis. (b) Verification of the alignment of the error bounds of both algorithms at $\varepsilon = 1.5$.

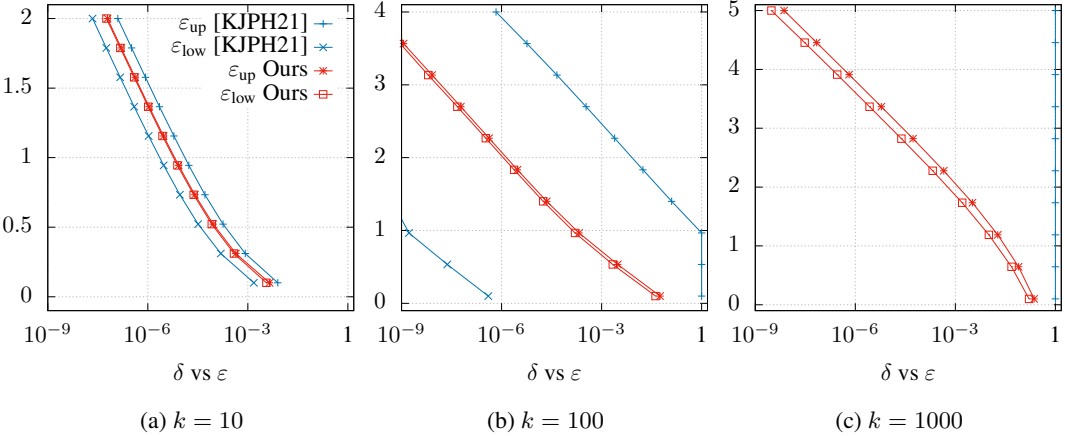

Figure 5: Comparing different error bounds using the same mesh size $8 \times 10^{-4}$ under different number of steps $k = 10, 100, 1000$. (With $p = 10^{-2}$, $\sigma = 0.8$.)

The accuracy of different algorithms are shown when a fixed mesh size is chosen in Figure 5. While for a small number of compositions, the algorithm of [KJPH21] gives reasonable estimates, for a large number of compositions, their error bounds worsen quickly.

---

[8]We are using the implementation of [KJPH21] from [KP21].

## B    Effect of floating point arithmetic

In this section, we demonstrate the effect of floating point inaccuracies on the computed privacy parameters. Figure 6 compares lower and upper bounds of the privacy curve with the analytical solution for small values of $\delta$. Our implementation uses `long double` floating point format which is platform dependent, however, it guarantees a precision at least as good as double precision which has a resolution of $10^{-15}$. The number of discretization points in this examples are on the order of $10^4$. Consequently, we expect floating point inaccuracies to become dominant for values on the order of $10^{-11}$. This can be also seen in the illustration, where the lower and upper bound fail to produce meaningful results for $\delta < 2 \times 10^{-11}$. Our implementation therefore only allows $\delta$ values which are greater than $10^{-10}$ which suffices for practical use cases.

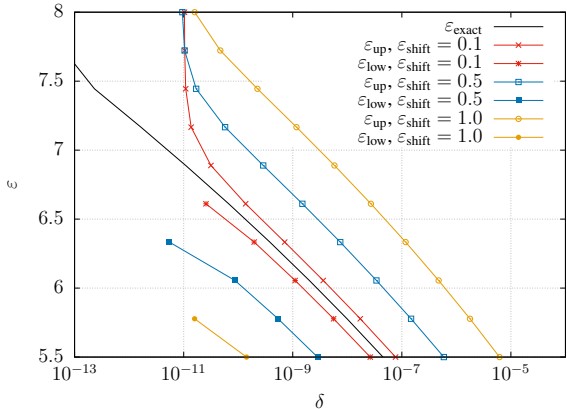

Figure 6: Setting $p = 1$ and comparing to the analytical solution (10) for values of $\delta$ beyond expected floating point accuracy.

## C    Privacy Loss Random Variables

In this section, we continue the discussion on privacy random variables in Section 3. First, we give the proof of the formula for PRVs of $\delta(P||Q)$ and the formula for a privacy curve given its PRVs (Theorem 3.2).

**Theorem 3.2.** *Let $\delta : \mathbb{R} \to [0, 1]$ be a privacy curve given by $\delta \equiv \delta(P||Q)$ where $P, Q$ are two random variables supported on $\Omega$. The PRVs $(X, Y)$ for the privacy curve $\delta$ are given by:*

$$X = \log\left(\frac{Q(\omega)}{P(\omega)}\right) \ \text{where} \ \omega \sim P,$$

$$Y = \log\left(\frac{Q(\omega)}{P(\omega)}\right) \ \text{where} \ \omega \sim Q.$$

*Proof.* We will first verify that $Y(t) = e^t X(t)$. This is equivalent to proving that $\mathbb{E}_Y[\phi(Y)] = \mathbb{E}_X[\phi(X)e^X]$ for any test function $\phi : \overline{\mathbb{R}} \to [0, 1]$. This is true since

$$\mathbb{E}_Y[\phi(Y)] = \mathbb{E}_{\omega \sim Q}\left[\phi\left(\log\left(\frac{Q(\omega)}{P(\omega)}\right)\right)\right]$$

$$= \mathbb{E}_{\omega \sim P}\left[\phi\left(\log\left(\frac{Q(\omega)}{P(\omega)}\right)\right)\frac{Q(\omega)}{P(\omega)}\right]$$

$$= \mathbb{E}_X[\phi(X)e^X].$$

We will now prove that $\delta(X||Y) = \delta(P||Q)$. We have

$$\delta(P||Q)(\varepsilon) = \sup_{S \subset \Omega} \Pr[Q \in S] - e^\varepsilon \Pr[P \in S]$$

$$= \Pr[Q \in S_\varepsilon] - e^\varepsilon \Pr[P \in S_\varepsilon]$$

where
$$S_\varepsilon = \left\{ \omega \in \Omega : \frac{Q(\omega)}{P(\omega)} > e^\varepsilon \right\} = \left\{ \omega \in \Omega : \log\left(\frac{Q(\omega)}{P(\omega)}\right) > \varepsilon \right\}.$$

Therefore $\Pr[Q \in S_\varepsilon] = \Pr[Y > \varepsilon]$ and $\Pr[P \in S_\varepsilon] = \Pr[X > \varepsilon]$. To complete the proof, note that

$$\delta(X||Y)(\varepsilon) = \sup_{T \subset \mathbb{R}} \Pr[Y \in T] - e^\varepsilon \Pr[X \in T]$$
$$= \Pr[Y \in T_\varepsilon] - e^\varepsilon \Pr[X \in T_\varepsilon]$$

where
$$T_\varepsilon = \left\{ t \in \overline{\mathbb{R}} : \frac{Y(t)}{X(t)} > e^\varepsilon \right\} = \left\{ t \in \overline{\mathbb{R}} : e^t > e^\varepsilon \right\} = (\varepsilon, \infty].$$

Putting it all together, we have:

$$\delta(P||Q)(\varepsilon) = \Pr[Y > \varepsilon] - e^\varepsilon \Pr[X > \varepsilon] = \delta(X||Y).$$

$\square$

**Theorem 3.3.** *The privacy curve $\delta$ can be expressed in terms of PRVs $(X, Y)$ as:*

$$\delta(\varepsilon) = \Pr[Y > \varepsilon] - e^\varepsilon \Pr[X > \varepsilon] = \mathbb{E}_Y[(1 - e^{\varepsilon - Y})_+] = \Pr[Y \geq \varepsilon + Z]. \quad (5)$$

*where $Z$ is an exponential random variable. Conversely, given a privacy curve $\delta : \mathbb{R} \to [0, 1]$, we can compute the PDFs of its PRVs $(X, Y)$ as:*

$$Y(t) = \delta''(t) - \delta'(t) \text{ and } X(t) = e^t(\delta''(t) - \delta'(t)). \quad (6)$$

*Proof.* Since the PDFs of PRVs $(X, Y)$ satisfy the relation $Y(t) = e^t X(t)$, we can rewrite the equation 5 in terms of just $Y$ or just $X$.

$$\delta(\varepsilon) = \Pr[Y \geq \varepsilon] - e^\varepsilon \Pr[X \geq \varepsilon]$$
$$= \int_\varepsilon^\infty Y(t)dt - \int_\varepsilon^\infty e^\varepsilon X(t)dt$$
$$= \int_\varepsilon^\infty Y(t)dt - \int_\varepsilon^\infty e^{\varepsilon - t} Y(t)dt \qquad \text{(Since } Y(t) = e^t X(t))$$
$$= \int_\varepsilon^\infty Y(t)(1 - e^{\varepsilon - t})dt$$
$$= \int_{-\infty}^\infty Y(t)(1 - e^{\varepsilon - t})_+ dt$$
$$= \mathbb{E}_Y[(1 - e^{\varepsilon - Y})_+]$$

To get the other form for $\delta(\varepsilon)$, we use the integration by parts formula.

$$\delta(\varepsilon) = \int_\varepsilon^\infty Y(t)(1 - e^{\varepsilon - t})dt$$
$$= \int_\varepsilon^\infty Y(t)dt + \int_\varepsilon^\infty (-Y(t)) e^{\varepsilon - t}dt$$
$$= \Pr[Y \geq \varepsilon] + \left( \Pr[Y \geq t]e^{\varepsilon - t} \Big|_\varepsilon^\infty - \int_\varepsilon^\infty \Pr[Y \geq t] \left(-e^{\varepsilon - t}\right) dt \right)$$
$$= \Pr[Y \geq \varepsilon] - \Pr[Y \geq \varepsilon] + \int_\varepsilon^\infty \Pr[Y \geq t]e^{\varepsilon - t}dt$$
$$= \int_\varepsilon^\infty e^{\varepsilon - t} \Pr[Y \geq t]dt$$
$$= \int_0^\infty e^{-z} \Pr[Y \geq \varepsilon + z]dz \qquad \text{(Substituting } z = t - \varepsilon)$$
$$= \Pr[Y \geq \varepsilon + Z]. \qquad \text{(where } Z \text{ is an exponential random variable)}$$

We now prove the converse relation by differentiating the expression for $\delta(\varepsilon)$ twice. We have:

$$\delta(\varepsilon) = \int_\varepsilon^\infty Y(t)dt - e^\varepsilon \int_\varepsilon^\infty e^{-t}Y(t)dt$$

$$\implies \delta'(\varepsilon) = -Y(\varepsilon) + e^\varepsilon \cdot e^{-\varepsilon}Y(\varepsilon) - e^\varepsilon \cdot \int_\varepsilon^\infty e^{-t}Y(t)dt = -e^\varepsilon \cdot \int_\varepsilon^\infty e^{-t}Y(t)dt$$

$$\implies e^{-\varepsilon}\delta'(\varepsilon) = -\int_\varepsilon^\infty e^{-t}Y(t)dt$$

$$\implies e^{-\varepsilon}\delta''(\varepsilon) - e^{-\varepsilon}\delta'(\varepsilon) = e^{-\varepsilon}Y(\varepsilon)$$

$$\implies Y(\varepsilon) = \delta''(\varepsilon) - \delta'(\varepsilon).$$

$\square$

## C.1 Examples of privacy loss random variables

In this section, we state the PRVs for a few standard mechanisms.

**Proposition C.1** (Gaussian Mechanism). *The PRVs for $\delta(\mathcal{N}(\mu,1)||\mathcal{N}(0,1))$ are:*

$$X = \mathcal{N}(-\mu^2/2, \mu^2) \text{ and } Y = \mathcal{N}(\mu^2/2, \mu^2).$$

*Proof.* Let $P = \mathcal{N}(\mu,1)$ and $Q = \mathcal{N}(0,1)$. By Theorem 3.2,

$$Y \sim \log\left(\frac{Q(t)}{P(t)}\right) \text{ where } t \sim Q$$

$$\sim \log\left(\frac{\exp(-t^2/2)}{\exp(-(t-\mu)^2/2)}\right) \text{ where } t \sim \mathcal{N}(0,1)$$

$$\sim \frac{(t-\mu)^2}{2} - \frac{t^2}{2} \text{ where } t \sim \mathcal{N}(0,1)$$

$$\sim \frac{\mu^2}{2} - \mu t \text{ where } t \sim \mathcal{N}(0,1)$$

$$= \mathcal{N}\left(\frac{\mu^2}{2}, \mu^2\right).$$

A similar calculation shows that $X = \mathcal{N}\left(-\frac{\mu^2}{2}, \mu^2\right)$ $\square$

**Proposition C.2** (Laplace Mechanism). *The PRVs for the privacy curve $\delta(\mathsf{Lap}(\mu,1)||\mathsf{Lap}(0,1))$ are:*

$$X = |Z| - |Z - \mu| \text{ and } Y = |Z - \mu| - |Z|$$

*where $Z \sim \mathsf{Lap}(0,1)$.*

*Proof.* Let $P = \mathsf{Lap}(\mu,1)$ and $Q = \mathsf{Lap}(0,1)$. By Theorem 3.2,

$$Y \sim \log\left(\frac{Q(t)}{P(t)}\right) \text{ where } t \sim Q$$

$$\sim \log\left(\frac{\exp(-|t|)}{\exp(-|t-\mu|)}\right) \text{ where } t \sim \mathsf{Lap}(0,1)$$

$$\sim |t-\mu| - |t| \text{ where } t \sim \mathsf{Lap}(0,1)$$

$$= |Z - \mu| - |Z| \text{ where } Z \sim \mathsf{Lap}(0,1).$$

A similar calculation shows that $X = |Z| - |Z - \mu|$ where $Z \sim \mathsf{Lap}(0,1)$. $\square$

**Proposition C.3** (($\varepsilon, \delta$)-DP). *The PRVs for the privacy curve of a $(\varepsilon, \delta)$-DP algorithm are*

$$X = \begin{cases} -\infty & \text{w.p. } \delta \\ -\varepsilon & \text{w.p. } \frac{(1-\delta)e^\varepsilon}{e^\varepsilon + 1} \\ \varepsilon & \text{w.p. } \frac{1-\delta}{e^\varepsilon + 1}, \end{cases}$$

$$Y = \begin{cases} -\varepsilon & \text{w.p. } \frac{1-\delta}{e^\varepsilon+1} \\ \varepsilon & \text{w.p. } \frac{(1-\delta)e^\varepsilon}{e^\varepsilon+1} \\ \infty & \text{w.p. } \delta. \end{cases}$$

*Proof.* It is easy to verify that $Y(t) = e^t X(t)$ for all $t \in \mathbb{R}$. We can also verify that
$$\delta(\varepsilon) = \Pr[Y > \varepsilon] - e^\varepsilon \Pr[X > \varepsilon] = \delta.$$
Morever $X = -Y$, therefore the privacy curve $\delta(X\|Y)$ is symmetric by Proposition D.9, i.e., $\delta(X\|Y) = \delta(Y\|X)$. These conditions together imply that $X, Y$ are PRVs for the $(\varepsilon, \delta)$-DP curve. $\qquad\square$

Note that in the all the above examples, we have $X = -Y$ as the privacy curves are symmetric.

## C.2 Subsampling

In this section, we calculate the PRVs for a subsampled mechanism given the PRVs for the original mechanism. Given two random variables $P, Q$ and a sampling probability $p \in [0, 1]$, $p \cdot P + (1-p) \cdot Q$ denotes the mixture where we sample $P$ w.p. $p$ and $Q$ w.p. $1 - p$.

**Proposition C.4.** *Let $(X, Y)$ be the PRVs for a privacy curve $\delta(P\|Q)$. Let $(X_p, Y_p)$ be the PRVs for $\delta_p = \delta(P\| p \cdot P + (1-p) \cdot Q)$. Then*
$$X_p = \log(1 + p(e^X - 1)),$$
$$Y_p = \begin{cases} \log(1 + p(e^Y - 1)) \text{ w.p. } p \\ \log(1 + p(e^X - 1)) \text{ w.p. } 1 - p. \end{cases}$$
*The CDFs of $X_p$ and $Y_p$ are given by:*
$$\mathrm{CDF}_{X_p}(t) = \begin{cases} \mathrm{CDF}_X\left(\log\left(\frac{e^t - (1-p)}{p}\right)\right) & \text{if } t \geq \log(1-p) \\ 0 & \text{if } t < \log(1-p) \end{cases}$$
$$\mathrm{CDF}_{Y_p}(t) = \begin{cases} p \cdot \mathrm{CDF}_Y\left(\log\left(\frac{e^t - (1-p)}{p}\right)\right) + (1-p) \cdot \mathrm{CDF}_X\left(\log\left(\frac{e^t - (1-p)}{p}\right)\right) & \text{if } t \geq \log(1-p) \\ 0 & \text{if } t < \log(1-p). \end{cases}$$

*Proof.* By Theorem 3.2,
$$X_p = \log\left(\frac{pY(t) + (1-p)X(t)}{X(t)}\right) \text{ where } t \sim X$$
$$= \log\left(pe^t + 1 - p\right) \text{ where } t \sim X$$
$$= \log\left(pe^X + 1 - p\right).$$
Similarly,
$$Y_p = \log\left(\frac{pY(t) + (1-p)X(t)}{X(t)}\right) \text{ where } t \sim pY + (1-p)X$$
$$= \log\left(pe^t + 1 - p\right) \text{ where } t \sim pY + (1-p)X$$
$$= \begin{cases} \log(1 + p(e^Y - 1)) \text{ w.p. } p \\ \log(1 + p(e^X - 1)) \text{ w.p. } 1 - p. \end{cases}$$
The CDF of $X_p$ is given by:
$$\Pr[X_p \leq t] = \Pr\left[\log\left(pe^X + 1 - p\right) \leq t\right]$$
$$= \Pr\left[X \leq \log\left(\frac{e^t - (1-p)}{p}\right)\right]$$
The CDF of $Y_p$ is given by:
$$\Pr[Y_p \leq t] = p \Pr\left[\log\left(pe^Y + 1 - p\right) \leq t\right] + (1-p) \Pr\left[\log\left(pe^X + 1 - p\right) \leq t\right]$$
$$= p \Pr\left[Y \leq \log\left(\frac{e^t - (1-p)}{p}\right)\right] + (1-p) \Pr\left[X \leq \log\left(\frac{e^t - (1-p)}{p}\right)\right].$$
$$\qquad\square$$

# D Missing Proofs in Error Analysis

## D.1 Facts about Coupling Approximation

Here we collect some useful properties of coupling approximations. The following lemma shows that the coupling approximations satisfy a triangle inequality.

**Lemma D.1** (Triangle inequality for couplings). *Suppose $X, Y, Z$ are random variables such that $|X - Y| \leq_{\eta_1} h_1$ and $|Y - Z| \leq_{\eta_2} h_2$. Then $|X - Z| \leq_{\eta_1 + \eta_2} h_1 + h_2$.*

*Proof.* There exists couplings $(X, Y)$ and $(Y, Z)$ such that

$$\Pr[|X - Y| \geq h_1] \leq \eta_1 \text{ and } \Pr[|Y - Z| \geq h_2] \leq \eta_2.$$

From these two couplings, we can construct a coupling between $(X, Z)$: sample $X$, sample $Y$ from $Y|X$ (given by coupling $(X, Y)$) and finally sample $Z$ from $Z|Y$ (given by coupling $(Y, Z)$). Therefore for this coupling, we have:

$$\begin{aligned}
\Pr[|X - Z| \geq h_1 + h_2] &\leq \Pr[|(X - Y) + (Y - Z)| \geq h_1 + h_2] \\
&\leq \Pr[|X - Y| + |Y - Z| \geq h_1 + h_2] \\
&\leq \Pr[|X - Y| \geq h_1] + \Pr[|Y - Z| \geq h_2] \\
&\leq \eta_1 + \eta_2.
\end{aligned}$$

$\square$

The following lemma shows that small total variation distance implies good coupling approximation.

**Lemma D.2.** *If the total variation distance $d_{TV}(X, Y) \leq \eta$, then $|X - Y| \leq_\eta 0$.*

*Proof.* It is well known that for any two random variables $X, Y$, there exists a coupling such that $d_{TV}(X, Y) = \Pr[X \neq Y]$. This immediately implies what we want. $\square$

## D.2 Bounding the error using tail bounds of PRVs

The goal of this section is to bound the error of ComposePRV in terms of the tail bounds of the underlying PRVs.

**Theorem D.3.** *Let $Y_1, Y_2, \ldots, Y_k$ be PRVs and let $\widetilde{Y}$ be the approximation produced by the **ComposePRV** algorithm (Algorithm 1) for $Y = \sum_{i=1}^{k} Y_i$ with truncation parameter $L$ and mesh size*

$$h = \frac{\varepsilon_{\text{error}}}{\sqrt{\frac{k}{2} \log \frac{2}{\eta_0}}}.$$

*Then*

$$\delta_{\widetilde{Y}}(\varepsilon + \varepsilon_{\text{error}}) - \delta_{\text{error}} \leq \delta_Y(\varepsilon) \leq \delta_{\widetilde{Y}}(\varepsilon - \varepsilon_{\text{error}}) + \delta_{\text{error}}$$

*where*

$$\begin{aligned}
\delta_{\text{error}} &= \Pr\left[\left|\sum_{i=1}^{k} \widetilde{Y}_i\right| \geq L\right] + \sum_{i=1}^{k} \Pr[|Y_i| \geq L] + \eta_0 \\
&\leq \Pr\left[\left|\sum_{i=1}^{k} Y_i\right| \geq L - \varepsilon_{\text{error}}\right] + 2\sum_{i=1}^{k} \Pr[|Y_i| \geq L] + 2\eta_0.
\end{aligned}$$

**Remark D.4.** *We can directly bound $\Pr\left[\left|\sum_{i=1}^{k} \widetilde{Y}_i\right| \geq L\right]$ using moment generating functions as*

$$\Pr\left[\left|\sum_{i=1}^{k} \widetilde{Y}_i\right| \geq L\right] \leq \inf_{\lambda > 0} \frac{\prod_{i=1}^{k} \mathbb{E}[\exp(\lambda \widetilde{Y}_i)]}{e^{\lambda L}} + \inf_{\lambda > 0} \frac{\prod_{i=1}^{k} \mathbb{E}[\exp(-\lambda \widetilde{Y}_i)]}{e^{\lambda L}}.$$

*Sometimes, if we already have good upper bound for $\Pr[|\sum_i Y_i| \geq L]$, then the second bound on $\delta_{\text{error}}$ is useful.*

The following key lemma shows that the DiscretizePRV algorithm (Algorithm 2) produces a good coupling approximation to the PRV and preserves the mean.

**Lemma D.5.** *Given a PRV $Y$, let $Y^L = Y\big|_{|Y| \leq L}$ be its truncation. The approximation $\widetilde{Y}$ returned by DiscretizePRV satisfies $\mathbb{E}[\widetilde{Y}] = \mathbb{E}[Y^L]$ and $|Y^L - (\widetilde{Y} - \mu)| \leq_0 \frac{h}{2}$ for some $\mu$ where $h$ is the mesh size. We also have $|Y^L - Y| \leq_\eta 0$ where $\eta = \Pr[|Y| \geq L]$.*

*Proof.* Since $d_{TV}(Y, Y^L) \leq \Pr[|Y| \geq L] = \eta$, by Lemma D.2, $|Y - Y^L| \leq_\eta 0$. It is clear that by construction $\mathbb{E}[\widetilde{Y}] = \mathbb{E}[Y^L]$,

$$\mathbb{E}[\widetilde{Y}] = \mu + \sum_{i=-n}^{n} ih \cdot q_i = \left(\mathbb{E}[Y^L] - \sum_{i=-n}^{n} ih \cdot q_i\right) + \sum_{i=-n}^{n} ih \cdot q_i = \mathbb{E}[Y^L].$$

We will now construct the coupling between $(Y^L, \widetilde{Y})$ such that $|Y^L - (\widetilde{Y} - \mu)| \leq \frac{h}{2}$. The coupling is as follows: First sample $y \sim Y^L$. Suppose $y \in (ih - \frac{h}{2}, ih + \frac{h}{2}]$ for some integer $i$ such that $-n \leq i \leq n$, then return $\widetilde{y} = \mu + ih$. Clearly, the distribution of $\widetilde{y}$ matches with $\widetilde{Y}$ and $|y - (\widetilde{y} - \mu)| = |y - ih| \leq \frac{h}{2}$.

$\square$

Since our error bound on $\widetilde{Y}$ is slightly different from the assumption in Lemma 5.3, we need the following generalization using the same proof.

**Lemma D.6.** *Suppose $Y_1, Y_2, \ldots, Y_k$ and $\widetilde{Y}_1, \widetilde{Y}_2, \ldots, \widetilde{Y}_k$ are two collections of independent random variables such that $|Y_i - (\widetilde{Y}_i - \mu_i)| \leq_0 h$ for some $\mu_i$ and $\mathbb{E}[Y_i] = \mathbb{E}[\widetilde{Y}_i]$ for all $i$, then*

$$\left|\sum_{i=1}^{k} Y_i - \sum_{i=1}^{k} \widetilde{Y}_i\right| \leq_\eta h\sqrt{2k \log \frac{2}{\eta}}.$$

In the algorithm, we only calculate the distribution of $Y_1 \oplus Y_2 \oplus \cdots \oplus Y_k$ instead of $Y_1 + Y_2 + \cdots + Y_k$. The following simple lemma shows that this is still a good approximation as long as $\sum_i Y_i$ stays within $[-L, L]$ with high probability.

**Lemma D.7.** *Let $Y_1, Y_2, \ldots, Y_k$ be random variables supported on $(-L, L)$. Then*

$$\left|\sum_{i=1}^{k} Y_i - (Y_1 \oplus_L Y_2 \oplus_L \cdots \oplus_L Y_k)\right| \leq_\eta 0$$

*where*

$$\eta = \Pr\left[\left|\sum_{i=1}^{k} Y_i\right| \geq L\right].$$

*Proof.*

$$\Pr\left[\sum_{i=1}^{k} Y_i \neq (Y_1 \oplus_L Y_2 \oplus_L \cdots \oplus_L Y_k)\right] \leq \Pr\left[\left|\sum_{i=1}^{k} Y_i\right| \geq L\right].$$

This clearly implies what we want. $\square$

Combining all the above lemmas, we get the following corollary.

**Corollary D.8.** *Let $Y_1, Y_2, \ldots, Y_k$ be random variables supported on and let $\widetilde{Y}_i$ be the discretization of $Y_i$ produced by DiscretizePRV algorithm with mesh size $h = \frac{h_0}{\sqrt{\frac{k}{2} \log \frac{2}{\eta_0}}}$ and truncation parameter $L$. Then*

$$\left|(Y_1 + Y_2 + \cdots + Y_k) - (\widetilde{Y}_1 \oplus \widetilde{Y}_2 \oplus \cdots \oplus \widetilde{Y}_k)\right| \leq_\eta h_0$$

*where*

$$\eta = \Pr\left[\left|\sum_{i=1}^{k} \widetilde{Y}_i\right| \geq L\right] + \sum_{i=1}^{k} \Pr[|Y_i| \geq L] + \eta_0.$$

*Furthermore, we can bound*

$$\Pr\left[\left|\sum_{i=1}^{k}\widetilde{Y}_i\right| \geq L\right] \leq \Pr\left[\left|\sum_{i=1}^{k} Y_i\right| \geq L - h_0\right] + \sum_{i=1}^{k} \Pr[|Y_i| \geq L] + \eta_0.$$

*Proof.* Let $Y^L \equiv Y_i\big|_{|Y_i| \leq L}$ be the truncation of $Y_i$. By Lemma D.5, $|Y_i^L - (\widetilde{Y}_i - \mu_i)| \leq_0 \frac{h}{2}$ for some $\mu_i$ and $|Y_i^L - Y_i|_{\xi_i} \leq 0$ where $\xi_i = \Pr[|Y_i| \geq L]$. Now applying the triangle inequality for coupling approximations (Lemma D.1), we have

$$\left|\sum_i Y_i - \sum_i Y_i^L\right| \leq_{\eta_1} 0$$

where $\eta_1 = \sum_i \xi_i = \sum_i \Pr[|Y_i| \geq L]$. By Lemma D.6, we have

$$\left|\sum_i Y_i^L - \sum_i \widetilde{Y}_i\right| \leq_{\eta_0} \frac{h}{2} \cdot \sqrt{2k \log \frac{2}{\eta_0}} = h\sqrt{\frac{k}{2} \log \frac{2}{\eta_0}} = h_0.$$

By Lemma D.7,

$$\left|\sum_{i=1}^{k} \widetilde{Y}_i - \left(\widetilde{Y}_1 \oplus_L \widetilde{Y}_2 \oplus_L \cdots \oplus_L \widetilde{Y}_k\right)\right| \leq_{\eta_2} 0$$

where $\eta_2 = \Pr\left[\left|\sum_{i=1}^{k} \widetilde{Y}_i\right| \geq L\right]$. Finally applying triangle inequality (Lemma D.1) once again, we get:

$$\left|(Y_1 + Y_2 + \cdots + Y_k) - (\widetilde{Y}_1 \oplus_L \widetilde{Y}_2 \oplus_L \cdots \oplus_L \widetilde{Y}_k)\right| \leq_{\eta} h_0$$

where $\eta = \eta_0 + \eta_1 + \eta_2$. We can bound $\Pr\left[\left|\sum_{i=1}^{k}\widetilde{Y}_i\right| \geq L\right]$ as:

$$\Pr\left[\left|\sum_i \widetilde{Y}_i\right| \geq L\right] = \Pr\left[\left|\sum_i (\widetilde{Y}_i - Y_i^L) + \sum_i (Y_i^L - Y_i) + \sum_i Y_i\right| \geq L\right]$$

$$\leq \Pr\left[\left|\sum_i (\widetilde{Y}_i - Y_i^L)\right| + \left|\sum_i (Y_i^L - Y_i)\right| + \left|\sum_i Y_i\right| \geq h_0 + 0 + L - h_0\right]$$

$$\leq \Pr\left[\left|\sum_i (\widetilde{Y}_i - Y_i^L)\right| > h_0\right] + \Pr\left[\left|\sum_i (Y_i^L - Y_i)\right| > 0\right] + \Pr\left[\left|\sum_i Y_i\right| \geq L - h_0\right]$$

$$\leq \eta_0 + \eta_1 + \Pr\left[\left|\sum_{i=1}^{k} Y_i\right| \geq L - h_0\right].$$

$\square$

*Proof of Theorem D.3.* Combining Corollary D.8 (with $h_0 = \varepsilon_{\text{error}}$) and Lemma 5.2, we have Theorem D.3. $\square$

## D.3 Tail Bound for PRVs

To finish the proof of our main theorem (Theorem 5.5, we need a tail bound on PRVs in terms of their privacy curves. First, we need a lemma relating the PRVs of a privacy curve $\delta(P||Q)$ with the PRVs of $\delta(Q||P)$.

**Proposition D.9.** *Let $(X, Y)$ be the PRVs for a privacy curve $\delta(P||Q)$. Then the PRVs for the privacy curve $\delta(Q||P)$ are $(-Y, -X)$.*

*Proof.* Let $(\widetilde{X}, \widetilde{Y})$ be the PRVs for $\delta(Q||P)$. We know that $\delta(P||Q) = \delta(X||Y)$. So $\delta(Q||P) = \delta(Y||X)$. Then by Theorem 3.2,

$$\widetilde{X} = \log\left(\frac{X(t)}{Y(t)}\right) \text{ where } t \sim Y$$
$$= \log\left(e^{-t}\right) \text{ where } t \sim Y$$
$$= -Y.$$

$$\widetilde{Y} = \log\left(\frac{X(t)}{Y(t)}\right) \text{ where } t \sim X$$
$$= \log\left(e^{-t}\right) \text{ where } t \sim X$$
$$= -X.$$

$\square$

Now, we show our tail bound, which shows the PRVs $(X, Y)$ for a $(\varepsilon, \delta)$-DP algorithm satisfies roughly that $\Pr(|Y| \geq \varepsilon + 2) \leq 2\delta$.

**Lemma 5.4.** *Let $(X, Y)$ be the privacy random variables for a $(\varepsilon, \delta)$-DP algorithm, then for any $t \geq 0$, we have*

$$\Pr[|Y| \geq \varepsilon + t] \leq \frac{\delta\left(1 + e^{-\varepsilon - t}\right)}{1 - e^{-t}}.$$

*Proof.* We have $\delta(X||Y) \leq f_{\varepsilon,\delta}$ and $\delta(Y||X) \leq f_{\varepsilon,\delta}$ where $f_{\varepsilon,\delta}$ is the privacy curve of a $(\varepsilon, \delta)$-DP algorithm. By Theorem 3.2, we have

$$\delta \geq \int_0^\infty \Pr[Y \geq \varepsilon + s]e^{-s}ds$$
$$\geq \int_0^t \Pr[Y \geq \varepsilon + s]e^{-s}ds$$
$$\geq \Pr[Y \geq \varepsilon + t]\int_0^t e^{-s}ds$$
$$\geq \Pr[Y \geq \varepsilon + t](1 - e^{-t}).$$

By Proposition D.9, the PRVs for $\delta(Y||X)$ are $(-Y, -X)$. Therefore by a similar argument, we have

$$\Pr[X \leq -\varepsilon - t] = \Pr[-X \geq \varepsilon + t] \leq \frac{\delta}{1 - e^{-t}}.$$

Finally, note that $Y(s) = e^s X(s)$ for all $s \in \mathbb{R}$ and $Y(-\infty) = 0$ by the definition of PRVs. Therefore

$$\Pr[Y \leq -\varepsilon - t] \leq e^{-\varepsilon - t}\Pr[X \leq -\varepsilon - t].$$

Therefore we have:

$$\Pr[|Y| \geq \varepsilon + t] = \Pr[Y \geq \varepsilon + t] + \Pr[Y \leq -\varepsilon - t]$$
$$\leq \Pr[Y \geq \varepsilon + t] + e^{-\varepsilon - t}\Pr[X \leq -\varepsilon - t]$$
$$\leq \left(1 + e^{-\varepsilon - t}\right)\frac{\delta}{1 - e^{-t}}.$$

$\square$

### D.4  Proof of Theorem 5.5

Now, we can prove our main theorem.

**Theorem 5.5.** *Let $\varepsilon_{\text{error}}, \delta_{\text{error}} > 0$ be some fixed error terms. Let $\mathcal{M}_1, \mathcal{M}_2, \ldots, \mathcal{M}_k$ be DP algorithms with privacy curves $\delta_{\mathcal{M}_i}(\varepsilon)$. Let $Y_i$ be the PRV corresponding to $\mathcal{M}_i$ such that $\delta_{\mathcal{M}_i}(\varepsilon) = \delta_{Y_i}(\varepsilon)$ for $\varepsilon \geq 0$. Let $\mathcal{M}$ be the (adaptive) composition of $\mathcal{M}_1, \mathcal{M}_2, \ldots, \mathcal{M}_k$ and let $\delta_{\mathcal{M}}(\varepsilon)$ be its privacy curve. Set $L \geq 2 + \varepsilon_{\text{error}}$ sufficiently large such that*

$$\sum_{i=1}^{k} \delta_{\mathcal{M}_i}(L-2) \leq \frac{\delta_{\text{error}}}{8} \text{ and } \delta_{\mathcal{M}}(L-2-\varepsilon_{\text{error}}) \leq \frac{\delta_{\text{error}}}{4}. \tag{7}$$

*Let $\widetilde{Y}$ be the approximation of $Y = \sum_{i=1}^{k} Y_i$ produced by* **ComposePRV** *algorithm with mesh size*

$$h = \frac{\varepsilon_{\text{error}}}{\sqrt{\frac{k}{2} \log \frac{12}{\delta_{\text{error}}}}}.$$

*Then*

$$\delta_{\widetilde{Y}}(\varepsilon + \varepsilon_{\text{error}}) - \delta_{\text{error}} \leq \delta_Y(\varepsilon) = \delta_{\mathcal{M}}(\varepsilon) \leq \delta_{\widetilde{Y}}(\varepsilon - \varepsilon_{\text{error}}) + \delta_{\text{error}}. \tag{8}$$

*Furthermore, our algorithm takes $O\left(b\frac{L}{h}\log\left(\frac{L}{h}\right)\right)$ time where $b$ is the number of distinct algorithms among $\mathcal{M}_1, \mathcal{M}_2, \ldots, \mathcal{M}_k$.*

*Proof.* By Lemma 5.4,

$$\Pr[|Y_i| \geq L] = \Pr[|Y_i| \geq L - 2 + 2]$$
$$\leq \delta_{Y_i}(L-2) \cdot \frac{1+e^{-L}}{1-e^{-2}}$$
$$\leq \delta_{Y_i}(L-2) \cdot \frac{1+e^{-2}}{1-e^{-2}}$$
$$\leq \delta_{Y_i}(L-2) \cdot \frac{4}{3}.$$

Therefore we have

$$\sum_{i=1}^{k} \Pr[|Y_i| \geq L] \leq \frac{4}{3} \sum_{i=1}^{k} \delta_{Y_i}(L-2) \leq \frac{4}{3} \cdot \frac{\delta_{\text{error}}}{8} = \frac{\delta_{\text{error}}}{6}.$$

Similarly

$$\Pr[|Y| \geq L - \varepsilon_{\text{error}}] \leq \frac{4}{3}\delta_Y(L-2-\varepsilon_{\text{error}}) \leq \frac{\delta_{\text{error}}}{3}.$$

Therefore by Theorem D.3, setting $\eta_0 = \frac{\delta_{\text{error}}}{6}$, we get the desired result.

For the runtime, we note that the bottleneck of our algorithm is to compute the convolution, which can be done using FFT. In total, we need to compute $b+1$ many FFT for $b$ distinct algorithms, one for each for computing the Fourier transform and one of computing the inverse Fourier transform. Since the length of the array for the FFT is bounded by $O(L/h)$, this costs $O(bL/h \log(L/h))$ in total.

The step $\delta_{\widetilde{Y}}(\varepsilon) = \mathbb{E}_{\widetilde{Y}}\left[\left(1 - e^{\varepsilon-\widetilde{Y}}\right)_+\right]$ can be computed in linear time by first computing the CDF of $\widetilde{Y}$ and the prefix sum $\mathbb{E}_{\widetilde{Y} \leq \alpha}\left[e^{-\widetilde{Y}}\right]$ for all $\alpha$. $\qquad\square$