# OpenReview forum: "Numerical Composition of Differential Privacy"
_NeurIPS.cc/2021/Conference — NeurIPS 2021 Spotlight_

### Official Review · Reviewer_9iuY · 2021-07-16

**Rating:** 7
**Confidence:** 4

**Summary:**

The proposes a refined numerical method for (eps,delta)-differential privacy accounting. The method is based on that of [KJPH21]: the discrete privacy random variable (PRV) is formed by using the CDF of the true PRV and then normalised. By then using concentration inequalities, the error analysis is simplified (compared to [KJPH21]) which also leads to tighter bounds that enable choosing smaller parameter values for the algorithm. This improved error analysis also leads to improvement of the theoretical complexity bound (this is also considerably lower than the one given by Murtagh and Vadhan (2016)).

**Limitations And Societal Impact:**

Yes, relation to the other accounting methods is discussed. This is mostly a theory paper.

**Main Review:**

All in all I think this is a very nice addition to this line of research, and I am leaning towards accepting the paper. I do have some comments and criticism, in more detail below. I am ready to reconsider my score based on authors' response.

Pros:

- The paper completes the previously proposed numerical method: The idea of normalising the truncated distributions and resorting to tail bounds in the error analysis (the way it is done) feels like the right way to do the error analysis. It is difficult to imagine major improvements to the discretisation procedure and to the overall complexity bounds. The complexity bound is considerably  better than the result given by Murtagh and Vadhan (2016).

- The paper seems well written and the presentation of the error analysis elegant.

Cons:

- The nature of the contribution is slightly incremental when considering the existing numerical method by [KJPH21]: the novelties lie mainly in the discretisation of the privacy loss random variable and in the error analysis of the resulting approximation.

- The experiments: The 'competing method' is obviously that of [KJPH21]. However, the chosen experimental comparisons (e.g. Figure 1a, Figure 3) seem to be chosen such that the method of [KJPH21] is not yet converging properly, and therefore do not seem to tell the whole truth. The method of [KJPH21] in a sense already 'does the job' by giving accurate tight bounds in affordable runtimes (compute times orders of seconds). E.g. in Figure 3b, I would expect the method of [KJPH21] to give tighter bounds (quickly) when decreasing the discretisation parameter h from the reported $8 \cdot 10^{-4}$ to $1 \cdot 10^{-5}$.


Questions and Remarks:

- Could you explain where does the assumption "$| Y_i - \widetilde{Y}_i | \leq_0 h$" in Lemma 5.3 come from? I would imagine there is always some gap due to the normalisation carried out for the truncated distribution so that for small enough $h$, $Y_i$ and $\widetilde{Y}_i$ would actually differ with a positive probability. Unless the discretisation interval contains the whole of $Y$, due to the normalisation $Y_i$ and $\widetilde{Y}_i$ would always differ.

- To make everything rigorous, should you not consider discretisation at the points $\{i \cdot h\}$, $-n \leq i < n$ instead of $-n \leq i \leq n$, due to the periodisation ?

- How is the PRV evaluated for the subsampled Gaussian mechanism exactly? Looking at the discretisation algorithm, you use the CDF at ih - h/2 and at ih + h/2 to obtain the point mass.  So you numerically integrate the CDFs in high precision using the analytic formula for the PRV? How do you ensure that the integration error stays very small e.g. for the PRV of the subsampled Gaussian mechanism?

- Can you get the FFT evaluations down to order of tens of milliseconds with reasonable accuracy? Unless that is the case, the gains from the computational speed up might be limited in practice.

- Remark about the runtime experiments: using those parameter values ($\sigma=0.8$, $p=4 \cdot 10^{-3}$, $\varepsilon=1.5$), the $\delta$ goes up to ~ $3 \cdot 10^{-5}$ for $k=2000$, and up to ~0.02 for $k=10^4$, so I would say that for up to $k=10^4$ it is less typical DP-SGD training (unless very small amount of data, $\delta$ should be less than $1/N$, $N$ number of samples).


**Time Spent Reviewing:**

8

---

> ### Author Response · Authors · 2021-08-09
> **Author response to Reviewer 9iuY**
>
> Thanks for the review! The main contribution of our work is the improvement in runtime and memory as a function of $k$ from $k^{1.5}$ to $k^{0.5}$.
>
> Regarding runtime (and memory) comparison with the method of KJPH21, please see Figure 4 in Appendix B in the supplementary material. KJPH21 fails to run beyond $k=2000$ and the run time is growing rapidly from $k=100$ to $2000$. In contrast, our method runs in less than a second even when $k=10,000$. When $k=2000$, we already see 40x speedups. (By further optimizations to our code, we could improve this to get 100x speedups for $k=2000$.) We also note that the memory required also follows the same trajectory as the runtime. The runtime comparisons are done while the accuracy of both the algorithms are aligned (see Figure 5), so this is indeed a fair comparison. And finally, we want to note that while training large models like GPT3, one needs to train for $k\approx 300,000$ iterations. So really large values of k are actually useful in practice as well. In the next revision, we will move the runtime plot into the main body to make this point more clear.
>
> Q1: If $Y_i$ is a PRV, $Y_i^L=Y|_{|Y_i|\le L}$  be its truncation and $\tilde{Y}_i$ be the discretization of $Y_i^L$, then we have $|Y_i^L - \tilde{Y}_i|\le_0 h$. We will be applying Lemma 5.3 to conclude that $|\sum_i Y_i^L - \sum_i \tilde{Y}_i|\le_\eta h\sqrt{2k\log{2/\eta}}$. By using tail bounds on $Y_i$, we will conclude that $|\sum_i Y_i - \sum_i Y_i^L|\le_\eta 0$. Combining these two we get that $\sum_i \tilde{Y}_i$ is a good coupling approximation to $\sum_i Y_i$.
>
> Q2: You are correct, we can pad the pdf with a zero at the end to take care of this. We will comment on this in a future revision.
>
> Q3: We have an exact closed form for the CDF of PRV of subsampled Gaussian mechanism. See Proposition C.5 in the appendix. So there is no need for numerical integration to estimate $q_i$ in Algorithm 2. But to estimate $\mathbb{E}[Y^L]$ in Algorithm 2, we need numerical integration. But the integrand is piecewise smooth in our applications and it is known that one can get machine accuracy $\eta$ for integrating piecewise smooth functions in time $\log(1/\eta)$ using Gaussian quadrature. We also conducted validation experiments to make sure that if extend Figure 2 to really small values of $\delta$, the algorithm remains accurate until $\delta$ gets close to machine accuracy.
>
> Q4: We reduce the number of points in the discretization grid to $k^{0.5}$ from $k^{1.5}$ in the work of KJPH21. And the time for FFT and inverse FFT is proportional to the number of points in the discretization grid ($N\log N$ time if $N$ is the number of points). Therefore, we should expect, and we demonstrate experimentally, significant speedups.
>
> Q5: In a future revision, we will increase the $\sigma$ and decrease $p$ to get more reasonable $\delta$ for large values of $k$. Since our runtime mainly depends on $k, \epsilon_{error}, \delta_{error}$, this should not affect the speedups.

---

> > ### Comment · Reviewer_9iuY · 2021-08-16
> > **Reply to authors**
> >
> > Thanks for all the clarifications, especially for Q1!
> >
> > Regarding experimental comparisons: in a future revision also a figure comparing the convergence of the algorithm to that of KPJH21 as a function of number the discretisation points might be a good illustration of the benefits obtained from modifying the discretisation procedure.
> >
> > Small remark: In Definition 5.1, the sentence "...if there exists a coupling between Y1, Y2..." sounds a bit funny, as this is the definition of the coupling itself. E.g. "... if there exists eta>0 and h>0 such that..." would make it more consistent, in my opinion.

---

> > > ### Author Response · Authors · 2021-09-03
> > > **Thanks for the suggestions**
> > >
> > > Thanks for your suggestions!  In the next revision, we will add a plot of the number of discretization points vs convergence of the algorithm and compare it previous work.

---

### Official Review · Reviewer_MB3M · 2021-07-16

**Rating:** 9
**Confidence:** 5

**Summary:**

The paper studies numerical composition of DP from a computation complexity perspective. Previous FFT based algorithm is optimized in its selection of discretization grid and truncation threshold. The runtime is significantly improved from $k^{1.5}$ to $k^{0.5}$. It also show that the CLT approach can underreport $\varepsilon$ for very small $\delta$.

**Limitations And Societal Impact:**

The authors have adequately addressed the limitations and potential negative societal impact of their work.

**Main Review:**

Although the algorithm is basically the same as the previous work, the improvement brought by the correct selection of discretization grid and truncation threshold is very remarkable. This shows the exciting potential of the FFT type accountant.

The concept of coupling approximation used in the analysis also seems interesting.

I'd like to point out something that complements the knowledge. In this paper, Definition 3.1 and the claim below gives a clear picture about what (pair of) densities can be PRVs, but since the so-called privacy curve is a more fundamental object (indeed, the whole paper is about computing an operation of this object), it would be great if there is a clear picture about the privacy curve. This appears in the following paper

> Zhu, Yuqing, Jinshuo Dong, and Yu-Xiang Wang. "Optimal Accounting of Differential Privacy via Characteristic Function."

In the same spirit, from the claim below Definition 3.1, it seems one of the two densities determines the other. This seems worth a serious statement, and I wonder if we can pick a simpler object to work on. It's also not clear to me what privacy curve guarantees density and what privacy curve guarantees discrete probability masses. Is there no privacy curve that we ever imagine to design that has non-discrete PRVs but also no density?

Regarding the writing, for the ease of the most general audience, I think including the runtime plot would be very helpful. Due to the page limit, Section 5 may have to be reduced. I understand how exciting it is to see theoretical ideas lead to practical improvements and the authors would like to tell us why, but I guess by human nature, a plot is more informative.

**Time Spent Reviewing:**

1.5

---

> ### Author Response · Authors · 2021-08-09
> **Author response to Reviewer MB3M**
>
> Thanks for the positive review and thanks for pointing out ZDW paper, we will go through it and cite it in the next revision. We also want to point out that, in DRS19 paper (which we cite), they already gave a complete characterization of privacy curves, i.e., which functions $\delta:\mathbb{R} \to [0,1]$ can be privacy curves. We will make this clear in the next revision.
>
> Our algorithm works even when the PRV is neither discrete nor has a density. Please see the remarks below Definition 3.1. Algorithm 2 only needs access to the CDF of the PRV which is always well defined. If the privacy curve is smooth, the PRVs will have density.
>
> We added the runtime plot in the appendix for lack of space, but we agree with the reviewer and will move it into the main body in the next revision.

---

> > ### Comment · Reviewer_MB3M · 2021-08-16
> > **Still 9 but please make revisions in camera-ready**
> >
> > It wasn't clear to me what kind of object you consider as fundamental. If it was the privacy curve, then the construction of PRVs (whose CDFs are the input of your algorithms) from the privacy curve isn't clear in this paper. In other words, the first part of the whole algorithm is missing. There are a few examples in the appendix. The generic solution only appears in the remark of a proof in the appendix. It's also not clear how this is related to DRS19. Maybe these questions are clear to the authors, but I don't think it is clear to the readers. In other words, maybe the results are already there or very immediate, but the ZDW paper makes the picture clearer. In its current form I think many readers of this paper will wonder about these questions but fail to find the answer.
> >
> > Maybe I'm missing something, so I'd like to hear response, but even without a response I don't think it changes my evaluation (still a top 15% not just a top 50%), but I do recommend the authors make comments about these questions. The paper studies a very important question and has a remarkable answer. It ought to be understood by more people.

---

> > > ### Author Response · Authors · 2021-08-17
> > > **Thanks for the useful suggestions**
> > >
> > > You are right in asking the question about which object is considered fundamental. The privacy curve $\delta:\mathbb R \to [0,1]$, the tradeoff function $f:[0,1]\to [0,1]$ and the Privacy Random Variable (which is any random variable such that $\mathbb E[e^{-Y}]=1$) are all equivalent to each other. Alternatively, one can specify a privacy curve as $\delta \equiv \delta(P||Q)$ for two random variables. So depending on the context, we can either have access to the privacy curve or PRV $Y$ or distributions of $P,Q$. In this sense, we believe no single description is fundamental. We can easily move from one representation to the other ones. You are absolutely right that we didn't make these connections clear in the main body due to lack of space. In the main body, we talk about converting $\delta(P||Q)$ to PRV $Y$ and converting PRV $Y$ to a privacy curve $\delta$. The conversion from privacy curve $\delta$ to PRV $Y$ appears (implicitly) in Remark C.1 in the appendix as you rightly pointed out. We will move this to the main body as per your suggestion.
> > >
> > > We mentioned DRS19 ("Gaussian Differential Privacy") because it gave a simple and complete characterization of which functions $f:[0,1]\to [0,1]$ can be tradeoff functions. Since the privacy curve $\delta:\mathbb R \to [0,1]$ is related to tradeoff functions by convex duality, this also implies a characterization of which functions are privacy curves. Therefore we claimed that DRS19 gave a characterization of privacy curves. But you are right that ZDW21 paper actually executes this idea fully and gives a simple and explicit characterization of privacy curves. Since this paper appeared after our submission, we weren't aware of this. We will cite this in the future revision along with the other suggestions you made to improve clarity.

---

> > > > ### Comment · Reviewer_MB3M · 2021-08-17
> > > > **Thanks and I have no other concerns**
> > > >
> > > > Thank you for clarifying. I have no other concerns. Keep up the great work :-)

---

### Official Review · Reviewer_Vztb · 2021-07-17

**Rating:** 8
**Confidence:** 4

**Summary:**

The paper considers an important problem of computing the total privacy loss under composition. Since computing the exact (and optimal) privacy loss is computationally infeasible under complexity-theoretic assumptions, the paper computes an approximation.

**Limitations And Societal Impact:**

Yes

**Main Review:**

Except for spotting some typos (like trucation instead of truncation on page 19), I do not have any complaints about the paper. I only have positive feedback about the paper and I will note it down for the latter stages:
1. The paper studies a very important problem of computing an approximation to privacy loss under composition.
2. The paper is very well written, the proof is elegant and easy to follow.
3. The algorithm is very efficient (not just theoretically, but even practically).
4. The current submission can be easily plugged in any large-scale deployment. I stressed tested the algorithm and it performed really well.

**Time Spent Reviewing:**

7

---

> ### Author Response · Authors · 2021-08-09
> **Author response to Reviewer Vztb**
>
> Thanks for the positive review and for pointing out typos, we will correct them in the next revision.

---

### Official Review · Reviewer_Jo5A · 2021-07-18

**Rating:** 6
**Confidence:** 4

**Summary:**

The paper studies an interesting problem --- obtaining arbitrary tight privacy guarantees of composed mechanisms efficiently. On the one hand, the known efficient RDP-based moments accountant (O(1) computational time) cannot achieve an arbitrary accuracy. On the other hand, the Fast Fourier Transform (FFT) based approach provides explicit error bounds, but the computational complexity is O( k^1.5), where k is the number of compositions.
The paper provides an improved analysis for computing the privacy curve of the composition and saves a factor of k in the run time and memory.


**Limitations And Societal Impact:**

The potential negative societal impact is not applicable here.

**Main Review:**

Weak points.

The main results are quite limited delta with respect to previous work. Given that the paper's major contribution is to *speed up the privacy computation* for PLD-based accountants, I think this paper is marginally above the acceptance threshold despite its interesting viewpoint in the error analysis.

Strong points.

1. Compared to the prior work [KJPH21], the proposed algorithm shifts the discretized random variables and improves the approximation procedue using the coupling approximation.

2. They provide a tighter tail bound of the privacy loss random variables, which improves the truncation procedure and enables a wide choice on the mesh size ($1/\sqrt(k)$) compared to ($1/k$) in [KJPH21].


Question:

Does the paper support heterogeneous composition as that in [KH21]?

Minor comments:
The authors can improve the clarity by removing words like *in an influential paper* and *in an important paper*.


**Time Spent Reviewing:**

5 hours

---

> ### Author Response · Authors · 2021-08-09
> **Author response to Reviewer Jo5A**
>
> Thanks for the comments! Yes, the algorithm supports heterogeneous composition. See Theorem 1.5 in the paper.

---

### Decision · Program_Chairs · 2021-09-27

**Decision:**

Accept (Spotlight)

**Comment:**

The reviewers felt that, although the algorithm in this paper is very similar to previous work, the modifications feel like the "right" way to solve the problem, and the improvements are significant for an important problem. The reviewers agreed that this paper should be accepted as a spotlight.